**Characterizing the evolution of mass flow properties and dynamics through analysis of**
**seismic signals: Insights from the 18 March 2007 Mt. Ruapehu lake-breakout lahar**
Braden Walsh[1], Charline Lormand[2], Jon Procter[3], Glyn Williams-Jones[1]
[1]Department of Earth Sciences, Simon Fraser University, Burnaby, British Columbia, Canada
[2]Department of Earth Sciences, University of Durham, Durham, DH1 3LE, UK
[3]Volcanic Risk Solutions, Institute of Agriculture and Environment, Massey University,
Palmerston North, New Zealand
Corresponding Author: Braden Walsh (braden_walsh@sfu.ca)
**Abstract**
Monitoring for mass flows on volcanoes can be challenging due to the ever-changing landscape
along the flow path, which can drastically transform the properties and dynamics of the flow.
These changes to the flows require the need for detection strategies and risk assessment that
are tailored not only between different volcanoes, but at different distances along flow paths as
well. Being able to understand how a flow event may transform in time and space along the
channel is of utmost importance for hazard management. While visual observations and simple
measuring devices in the past have shown how volcanic mass flows transform along the flow
path, these same features for the most part have not been described using seismological
methods. On 18 March 2007, Mt. Ruapehu produced the biggest lahar in New Zealand in over
100 years. At 23:18 UTC the tephra dam holding the Crater Lake water back collapsed causing
$1.3 \times 10^6$ $m^3$ of water to flow out and rush down the Whangaehu channel. We describe here the
seismic signature of a lake-breakout lahar over the course of 83 km along the Whangaehu river
system using three 3-component broadband seismometers installed <10 m from the channel at
7.4, 28, and 83 km from the crater lake source. Examination of 3-component seismic
amplitudes, frequency content, and directionality, combined with video imagery and sediment
concentration data were used. The seismic data shows the evolution of the lahar as it
transformed from a highly turbulent out-burst flood (high peak frequency throughout), to a
fully bulked up multi-phase hyperconcentrated flow (varying frequency patterns depending on
the lahar phase) to a slurry flow (bedload dominant). Estimated directionality ratios show the
elongation of the lahar with distance down channel, where each recording station depicts a
similar pattern, but for differing lengths of time. Furthermore, using directionality ratios shows
extraordinary promise for lahar monitoring and detection systems where streamflow is present
in the channel.
**1. Introduction**
Volcanic mass flows (e.g. debris flows, pyroclastic density currents, debris avalanches,
hyperconcentrated flows) are one of the greatest threats to communities, industry, recreation,
etc. on and around volcanoes. Volcanic mass flows are particularly dangerous as they are fast
moving turbulent flows that can occur without any warning or an eruption transpiring (Capra et
al., 2010). These flows can move a sizable amount of liquid and debris great distances that can
critically impact locations hundreds of kilometers from the volcano or source. Lake-breakout or
outburst flood events can be particularly destructive because they tend to be larger and can
cause long lasting changes to the landscape and surrounding ecosystems (O'Connor et al., 2013;
Procter et al., 2021). Furthermore, unlike eruption or rain triggered mass flows, outburst floods
have very little to no warning. Eruption triggered flows can be prepared for by the onset of the
eruption and/or the monitoring of the volcano through various methods (e.g. seismology,
infrasound, gravity, gas and water chemistry). Likewise, for rain-induced flows using techniques
such as measuring the amount or intensity of rain (e.g. Capra et al., 2010; 2018) or by
monitoring the amount of available material (e.g. Iguchi, 2019) can help forecast when an event
may occur.
In New Zealand, there have been numerous cases of large damaging mass flows in modern
times. For example, in October 2012, a lake-breakout lahar originating from Te Maari,
destroyed hiking trails and forestry, eventually flowing over 4.5 km to damage and block off
Highway 46 (Procter et al., 2014; Walsh et al., 2016). Moreover, on 24 December 1953, the
deadliest volcanic mass flow in New Zealand history occurred killing 151 people when a lahar
struck a train crossing at the Tangiwai Rail Bridge, 39.8 km from the Crater Lake on top of Mt.
Ruapehu (O'Shea, 1954). The ability to predict and investigate the changing dynamics and
properties of large volcanic mass flows as they progress down channel is the first step in
beginning to understand flow mechanisms better, and ultimately address the hazards involved
to mitigate the risk.
In order to better characterize and understand these flow events, many in-situ applications and
instruments have been used in the past (e.g. trip wires, stage gauge, load cells, pore pressure).
While many of these tools can yield quick assessments and provide ample warning (e.g. current
meters, trip wires), they can sometimes be at risk of false detections, equipment damage or
loss, and/or lack the capability to evaluate multiple pulses or flow events (Arattano et al., 1999).
Geophysical instruments (e.g. seismometers, geophones, infrasound) on the other hand can be
installed at a safe distance away from the channel and have shown signs of not only being
capable warning systems (e.g. Coviello et al., 2019), but have the ability to accurately estimate
flow properties (e.g. Arattano and Marchi, 2005; Doyle et al., 2010; Schimmel et al., 2021), as
well as flow dynamics (e.g. Gimbert et al., 2014; Coviello et al., 2018; Walsh et al., 2020).
However, in order to fully utilize these instruments, improved interpretation, comprehension,
assessment, and universality is needed. One technique to increase the ability to predict, warn,
and estimate the properties and dynamics of flow events is to use all three components of the
seismic recording. Recently, several studies have shown that using all three components is
effective in characterizing flow events (e.g. snow-slurry lahars, Cole et al., 2009; snow
avalanches, Kogelnig et al., 2011; streamflow, Roth et al., 2016; landslides, Surinach et al., 2005;
lahars, Walsh et al., 2020; rockfalls, Kuehnert et al., 2021; hyperconcentrated flows, Walsh et
al., 2016). Using the horizontal components along with the vertical component can yield
additional information about the flow that is not utilized if only the vertical component is used.
Notably, directionality (cross-channel over channel-parallel) analysis (e.g. Doyle et al., 2010;
Walsh et al., 2020) can provide information about the wetted perimeter, sediment
concentration, and number of particle collisions. Furthermore, differing energies and frequency
outputs from channel parallel and channel perpendicular signals can point to specific changes
within the flow (Burtin et al., 2010; Roth et al., 2016) that can provide insights into the internal
dynamics.
**1.1 Anatomy of lahars**
When a lahar is created from a lake-breakout or outburst flood event, the transition from flood
or streamflow torrent depends on the erosivity of the channel and the supply of sediment being
entrained within the flow (e.g. Scott, 1988; Doyle et al., 2011). An event may start as a highly
turbulent low sediment flow, then transform into a hyperconcentrated flow, and may even
eventually 'bulk up' to exhibit characteristics of a debris flow with the possibility of plug-like
(limited internal motion and collisions) or laminar behavior (Scott, 1988, Pierson et al., 1990). At
Mt. Ruapehu, the propagational differences of lahars down channel have been observed and
characterized in the past (e.g. Cronin et al., 1996; Cronin et al., 1999; Cronin et al., 2000;
Manville et al., 2000; Procter et al., 2010a; Lube et al., 2012). From these studies, models of
how lahars bulk up and transition throughout the run-out distance have been postulated. For
the lahars in the Whangaehu channel, Cronin et al. (1999) created three 4-phase conceptual
models based on source distances of 23.5 km, 42 km, and >55 km. The first two models are for
lahar regimes, whereas the third model described a lahar almost at its peak run-out distance. In
each model, the first phase consists of a super charged streamflow pulse that flows ahead of
the head of the flow and is considered the front of the lahar. This phenomenon has also been
noted for debris flows interacting with streamflow (Arattano and Moia, 1999). Furthermore,
discharge is maximum at the transition between phase 1 and phase 2 (Cronin et al., 1999), and
is described as the head of the flow. Phase 2 is described as a mixing zone between streamflow
and increasing sediment content, where the peak sediment concentration usually occurs at the
end of phase 2 or at the beginning of phase 3 (e.g. Pierson and Scott, 1985). Cronin et al. (1999)
defined phase 3 as the lahar body, which has the least amount of the original streamflow
contained within. Phase 3 is also characterized by coarse sediment suspensions and is the most
likely location for debris flow rheology. Finally, phase 4 is the tail of the lahar where debulking
and dilution occurs transforming the lahar back into a hyperconcentrated, mixed, or
streamflow.
**1.2 18 March 2007 lake-breakout event**
Mt. Ruapehu (2797 asl) is the largest stratovolcano in the central North Island of New Zealand
(Figure 1) which sits at the southwestern end of the Taupō Volcanic Zone (TVZ). The volcano has
a volume of 110 km$^3$ which is composed of several overlapping cone building formations and
surrounding ring plain volcaniclastics (Carrivick et al., 2009; Pardo et al., 2012). On top of the
volcano, above the currently active vent sits a $1\times10^7$ m$^3$ acidic crater lake (Procter et al., 2010a).
The Whangaehu channel is the preferred outlet for Crater Lake water and lahars in recent
history (Procter et al., 2012; Procter et al., 2021). The Whangaehu channel is on the eastern
flank of Mt. Ruapehu where it runs down across the volcanic ring plane and eventually heads
southwest for ~200 km reaching the Tasman Sea (Figure 1).
Prior to the events that took place in the morning local time on 18 March 2007, a heavy
rainstorm occurred accumulating about 256 mm of water over the 10 hours prior to the dam
breach that led to the outburst flood (Massey et al., 2010). The intense rain caused the Crater
Lake to rise an extra 6.4 m and overtop the natural lava formation ledge, which started to cause
seepage and extra water to enter the Whangaehu gorge (Carrivick et al., 2009). At ~23:18 UTC,
the tephra dam collapsed causing $1.3\text{x}10^6$ m$^3$ of water to flow out of the lake and into the
Whangaehu channel (Procter et al., 2010a). The dam was eroded and undercut in multiple
stages resulting in a series of retrogressing landslides along with the main debris flow/lahar.

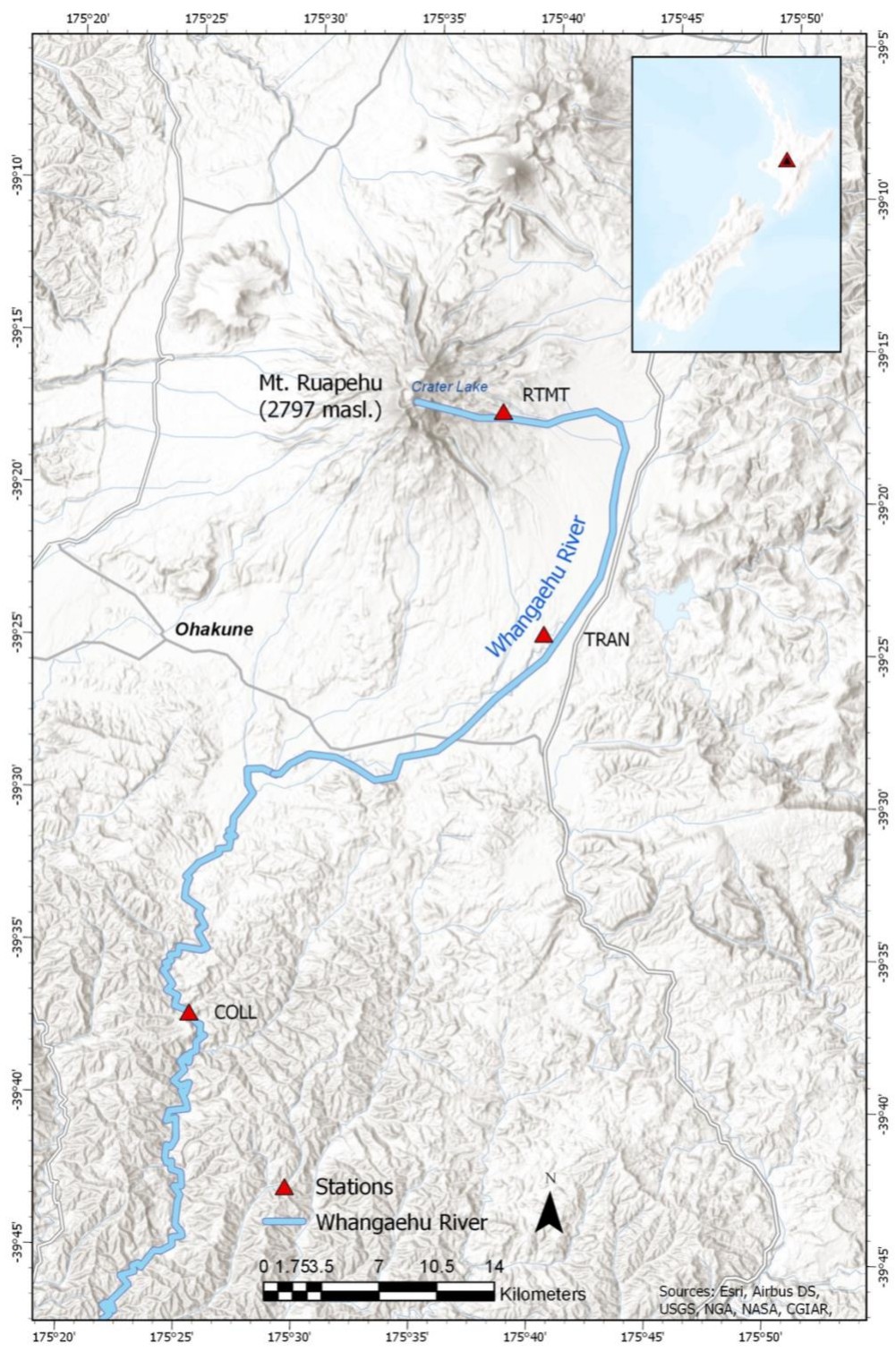


*Figure 1 Map of Mt. Ruapehu and the surrounding area located on the central North Island of New Zealand. Blue*
*outline represents the Whangaehu channel and the path the 18 March 2007 lahar traveled down. Red triangles*
*denotes the three monitoring stations along the Whangaehu channel at 7.4, 28, and 83 km.*

Since the lahar was caused by lake-breakout dynamics and thus contained an abundance of
water, the event was classified as a hyperconcentrated streamflow (Procter et al., 2010b). At
~8.0 km from source, the lahar velocity was recorded at ~ 9.5 m/s and had an estimated 6 m of
downcutting, showing the capability of the lahar to deposit and erode massive amounts of
material (Procter et al., 2010a,b). Furthermore, the 18 March 2007 lahar was one of the most
thoroughly monitored lahars ever (Manville and Cronin 2007). In total there were 21
monitoring locations (only three of which had 3-component seismometers) setup to measure
various lahar properties, (e.g. flow monitor, camera, stage height, flow sampling, pore-
pressure, seismic, etc.) along the channel (Keys and Green, 2008; Lube et al., 2012), with the
lahar taking over 16 hours to eventually travel out to the New Zealand coast, ~200 km from the
original crater lake source.
Here, we delve into the properties of the 18 March 2007 lake-breakout hyperconcentrated
streamflow that bulked up to a volume of ~$4.4 \times 10^6$ m$^3$ (Procter et al., 2010a) over the course of
83 km along the Whangaehu channel, originating from Mt. Ruapehu, New Zealand. The
combination of seismic analysis (frequency and directionality) with supplementary
measurements (e.g. video, sediment concentration) show how a lahar transforms over time and
distance and how using these seismic techniques can help monitor the ever changing dynamics
and properties of a flow event. Furthermore, we examine previous models of the evolution of a
lahar and compare the model with the seismic data available.
**2. Data**
The seismic data for the 18 March 2007 lahar was recorded on three seismometers installed at
various distances (7.4, 28, 83 km) along the Whangaehu channel (Figure 1). The data from the
three 3-component broadband Guralp 6T sensors (COLL, RTMT, TRAN) were recorded using a
sampling rate of 100 Hz and GPS time stamps. For each site, the seismometers axes were
installed to true North and the recorded data were rotated to align North as flow parallel (P)
and East as the cross-channel direction (T). The seismic data were rotated to align with the
channel in order to determine the differences in energy output between the flow parallel and
cross-channel directions (e.g. directionality). The monitoring station Round the Mountain Track
(RTMT), was installed 4 m from the channel and 7.4 km downstream from the source of the
lahar. The lahar arrived at RTMT at 23:36 UTC and had an average velocity of 9.3 m/s (Figure
2a). The Trans Rail Gauge (TRAN) station was installed 28 km from source and 10 m from the
channel, which also included a video camera that captured an image every 30 seconds. The
lahar arrived at TRAN at 24:35 UTC and had an average velocity of 5.6 m/s (Figure 2d). The
Colliers Bridge (COLL) station was installed 10 m from the channel and 83 km from source. The
lahar arrived at COLL at 04:13 UTC and had an average velocity of 4.8 m/s (Figure 2f). Arrival
times are based off of images and eye witnesses at each of the monitoring stations. The flow
velocity at RTMT and COLL were estimated from imagery and at TRAN from a flow meter.
Sediment concentration at COLL was measured manually through dip buckets.
**3. Results**
To examine the multi-component dynamics of the 18 March 2007 lake-breakout event along
the Whangaehu channel at three monitoring locations, the data were corrected for instrument
response and split into 10 s time windows. At each recording location, peak spectral frequency
(PSF), root mean squared (RMS) amplitude, and directionality ratios (DR) are estimated for each
of the 10 s time windows (Table S1). At each monitoring station the first hour of the lahar
including five minutes prior to the arrival are shown in all the results except when indicated.



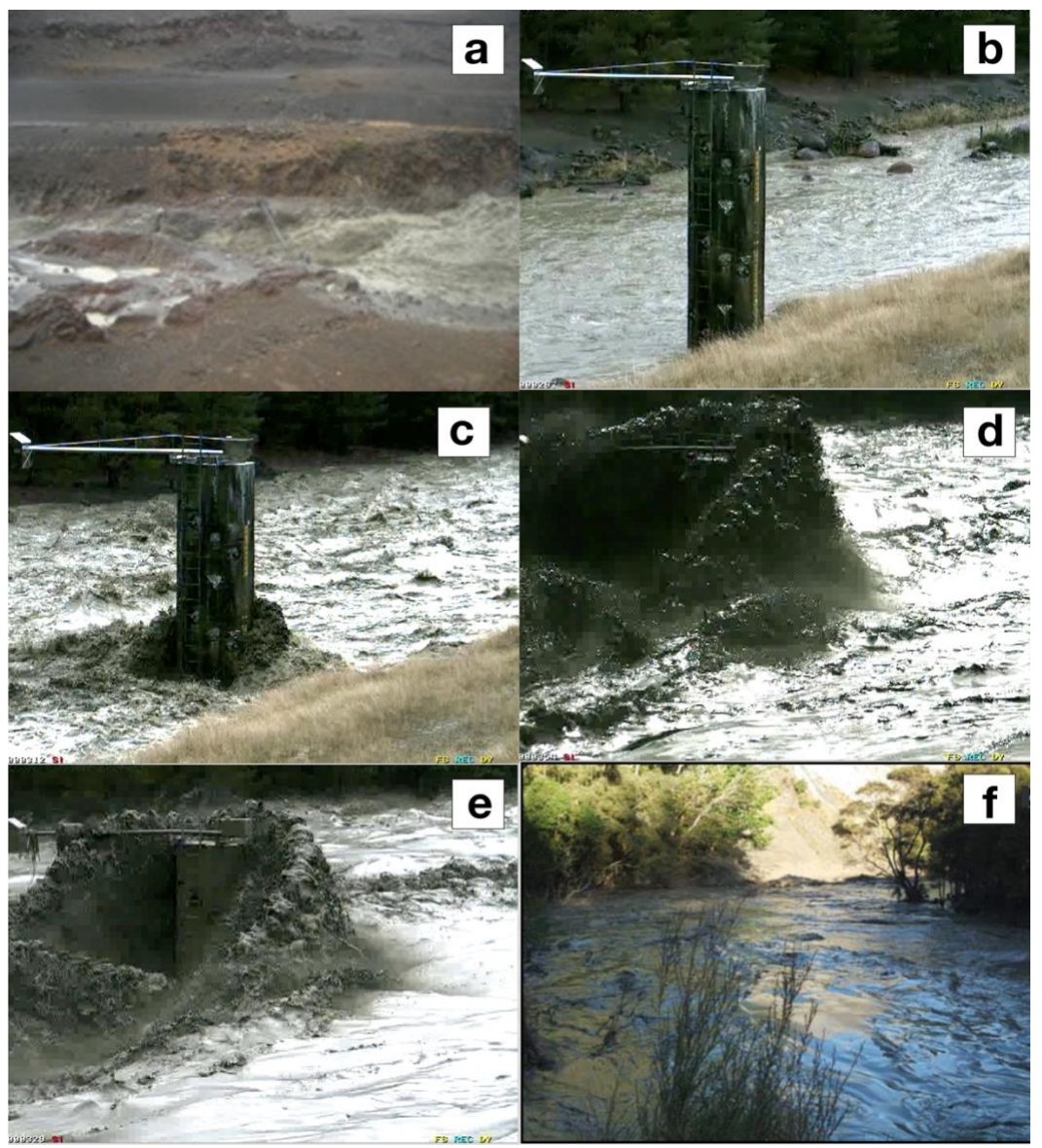

*Figure 2 Images from the 18 March 2007 lake break-out lahar from RTMT (a), TRAN (b, c, d, e), and COLL (f). Note*
*the transformation of the lahar at TRAN from streamflow (b), increased discharge pre-lahar phase 1 pulse/flow*
*front (bow wave) (c), head of the lahar (peak seismic amplitude) (d), and low PSF beginning of lahar body (see*
*figure 4 after 15 minute mark) (e).*

**3.1 Frequency analysis**

In order to examine the PSFs for all three components at each site along the channel, we use

the frequency recorded at the maximum amplitude of the frequency spectra for each 10 s

running time window (i.e. non-overlapping windowed FFT). The PSF for RTMT (7.4 km from

source) shows similar patterns between all three components (Figure 3). The five minutes prior
to the arrival of the front of the lahar are characterized by scattered PSFs between 20-40 Hz for
the cross-channel (Figure 3a) and parallel (Figure 3b) directions, while in the vertical direction
(Figure 3c) the PSF is ~30 Hz. When the front (streamflow pulse/bow wave) of the lahar arrives
at the station, the PSF in all three components decreases to ~5-10 Hz for about 1 min before
increasing again to higher frequencies. After the head (peak seismic amplitude) of the lahar
passes the station, the PSF in the cross-channel and parallel directions remain between 30-40
Hz for the rest of the recording window. In the vertical component, the PSF is scattered
between 20-40 Hz for ~15 min after the arrival of the head of the lahar and then becomes
narrower, similar to both the cross-channel and parallel components with PSFs between 30-40
Hz.

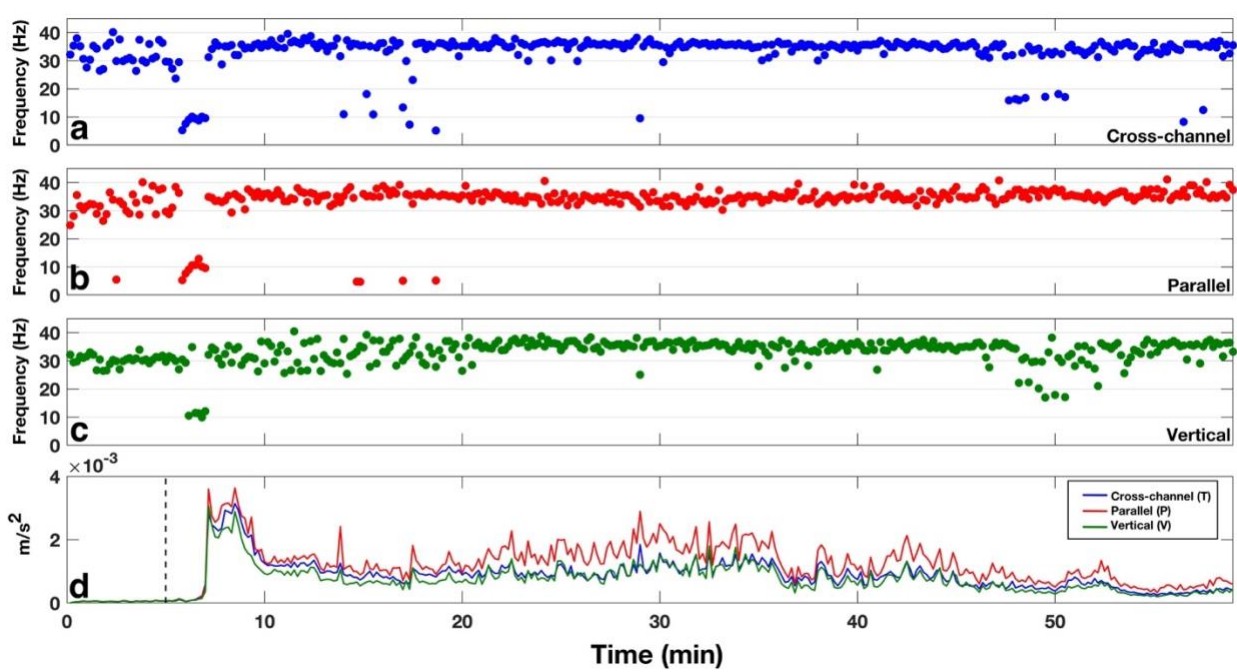


*Figure 3 Peak spectral frequencies for RTMT (7.4 km from source) for a) cross-channel (blue), b) flow parallel (red),*
*and c) vertical (gree) directions. Bottom row (d) depicts the RMS amplitude of the lahar color coded to the same*
Further down the channel at station TRAN (28 km from source), the PSFs for all three
components show a similar overall pattern (Figure 4). The pre-lahar PSF distribution in all three
components is between 20-32 Hz. Like RTMT higher up the channel, the PSFs for the front of
the lahar at TRAN drops down to around 10 Hz and when the lahar head arrives (~10 min,
Figure 4) the PSF increases to ~30 Hz for parallel (Figure 4b) and cross-channel (Figure 4a)
directions and between 20-30 Hz in the vertical component (Figure 4c). This decrease to lower
frequencies before the head of the lahar at TRAN lasts for about 5 min. After the head of the
lahar passes the recording station the PSF content decreases for ~15 min to 10-20 Hz for the
parallel and cross-channel components and between 10-25 Hz for the vertical components. The
PSF after the 30 minute mark in Figure 4 displays a bimodal pattern with frequencies between
10-35 Hz, with PSF time windows concentrating most at ~30 Hz.
At the COLL recording station (83 km from source), the PSF distribution shows differing patterns
for all three components (Figure 5). The PSF in the cross-channel direction (Figure 5a) depicts a
bimodal pattern throughout with a strong lower concentration of time windows at ~18 Hz and a
higher PSF at ~25 Hz. For the parallel component (Figure 5b), the pre-lahar signal has a wide PSF
range between 12-30 Hz. When the lahar arrives, the PSF becomes concentrated at ~22 Hz for
~8 min before transforming into a bimodal pattern similar to that of the cross-channel PSF, with
frequencies between 20-30 Hz.  In the vertical component (Figure 5c), the pre-lahar PSF is
scattered between 22-30 Hz, then as the front of the lahar passes the station, the PSF stabilizes
around 28 Hz for about 12 min. When the lahar head arrives, the PSF again transitions to more
of a scattered pattern during the highest energy stage of the lahar (Figure 5, 25-40 min).

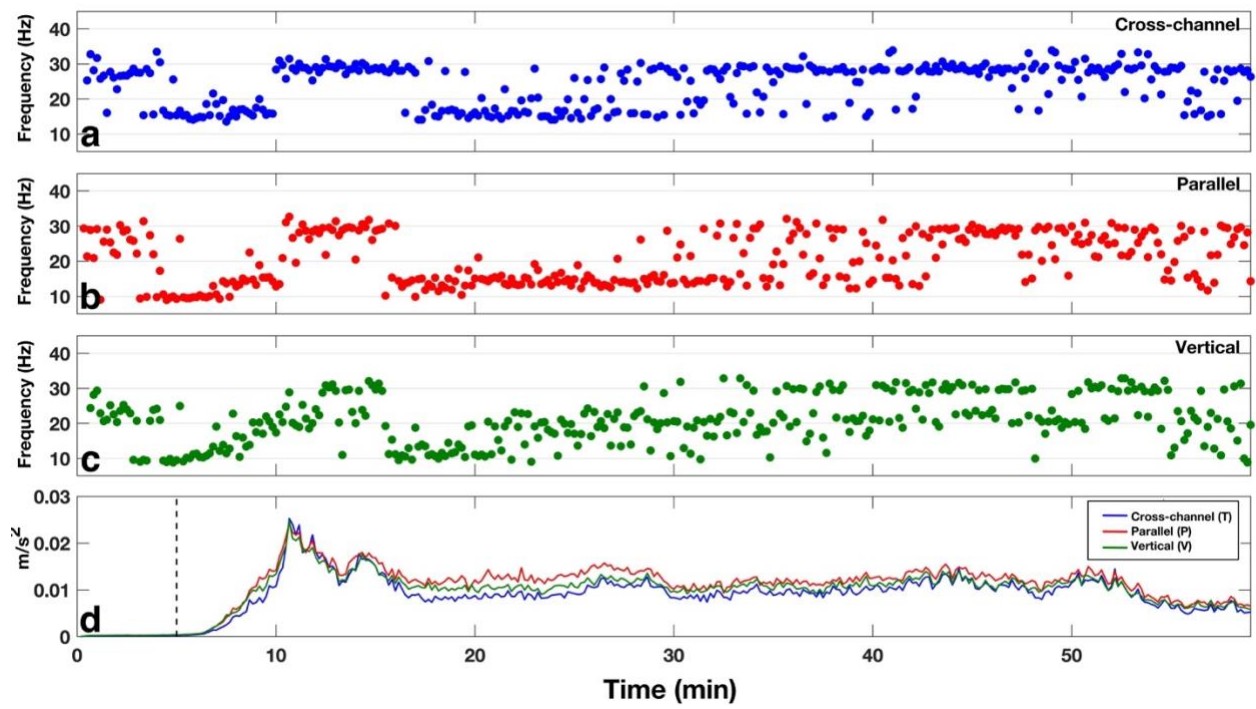


*Figure 4 Peak spectral frequencies for TRAN (28 km from source) for a) cross-channel (blue), b) flow parallel (red), and c) vertical (green) directions. Bottom row (d) depicts the RMS amplitude of the lahar color coded to the same colors as the PSF. The dashed vertical line marks the timing of the lahar front passing the monitoring station. All PSFs and RMS amplitudes were calculated using 10 s time windows.*

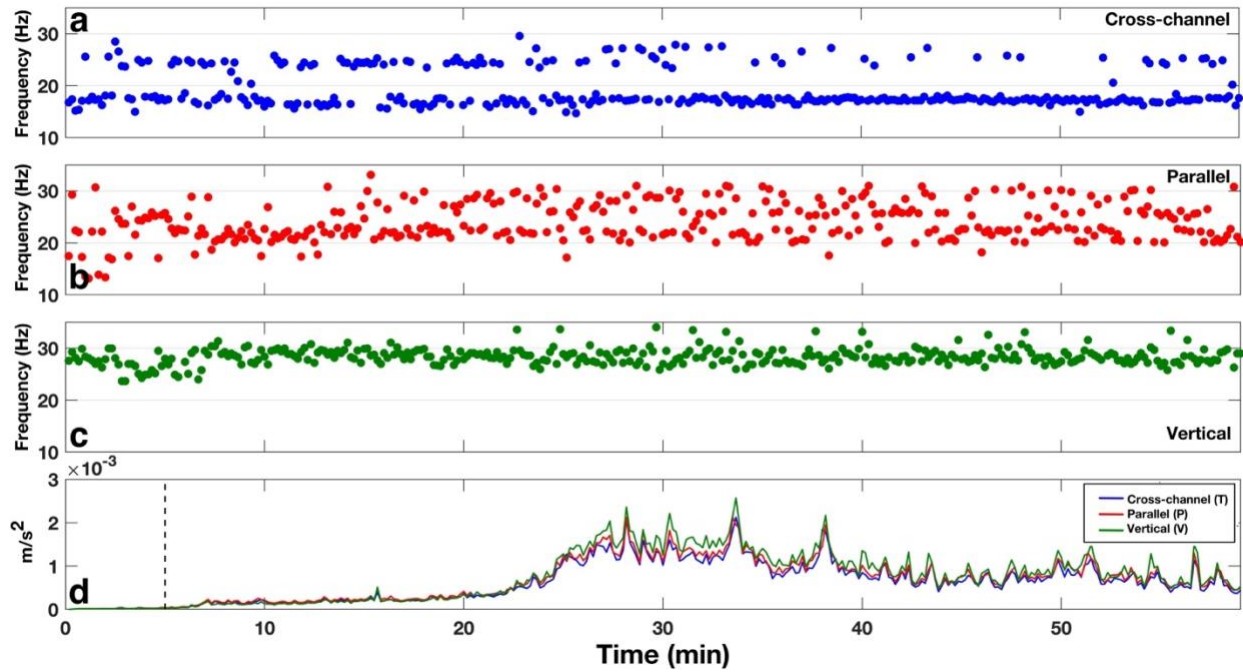


*Figure 5 Peak spectral frequencies for COLL (83 km from source) for a) cross-channel (blue), b) flow parallel (red),*
*and c) vertical (green) directions. Bottom row (d) depicts the RMS amplitude of the lahar color coded to the same*
*colors as the PSF. The dashed vertical line marks the timing of the lahar front passing the monitoring station. All*
*PSFs and RMS amplitudes were calculated using 10 s time windows.*
**3.2 Directionality**
When recording mass flows with 3-component sensors, the directionality may be examined due
to the sensor being able to record signals in the two horizontal directions. The directionality
ratio allows for the determination of which horizontal component has stronger energy over the
course of the recording window. This is possible because, in channel side deployments for mass
flow monitoring systems, the sensor is either installed so that the North component is aligned
to be parallel and the East component aligned as perpendicular to the flow, or the components
are rotated during the data processing stage to align with the channel orientation.
Furthermore, with the channel side installations, attenuational factors can mostly be ignored
due to the close proximity to the channel and energy output of the flow event. The
directionality ratio (DR) can be defined as the cross-channel amplitude divided by the flow
parallel amplitude. A DR > 1 indicates that the cross-channel amplitude is larger than that of the
flow parallel, and vice-versa for a DR < 1. Directionality ratios have been used in the past to
show rheology changes within flows, where the DR increases when streamflow transitions into
a lahar (Walsh et al., 2020), and have been hypothesized to be an indicator for flow properties
such as, sediment concentration, wetted perimeter, and/or amount of particle collisions within
a lahar (Doyle et al., 2010).
The directionality ratios estimated from 10 s non-overlapping running time windows of the RMS
amplitudes at each seismic station for the 18 March 2007 lake-breakout lahar are shown in
Figure 6. The DR for RTMT (Figure 6a) displays a DR <= 1 (0.8-1.0) pre-lahar, then decreases
(0.7-0.8) as the lahar arrives at the recording station (Figure 6, dashed line), then as soon as the
lahar head arrives, the DR increases to above DR = 1 for ~2 min. After the peak lahar flood pulse
passes RTMT, the DR then proceeds to decrease below a DR = 1 for the rest of the recording
window. Similar to RTMT, the DR for TRAN starts out with a DR < 1 (0.7-0.8) and as the lahar
front passes, the DR similarly decreases to 0.6-0.7 before increasing to a DR > 1 for ~5 min
when the lahar is at peak energy output starting at about the 10 min mark (Figure 6d, red line).
After the passing of the peak energy, the DR for TRAN decreases below 1 again for the
remainder of the recording window. Further down the channel at COLL (Figure 6c), the DR
before the lahar arrives has a wide range of values between 0.8-1.2. When the front of the
lahar passes (Figure 6, dashed line), the DR stabilizes between 0.8-1, before increasing slightly
when the peak energy of the lahar passes the monitoring site at about the 25 min mark.



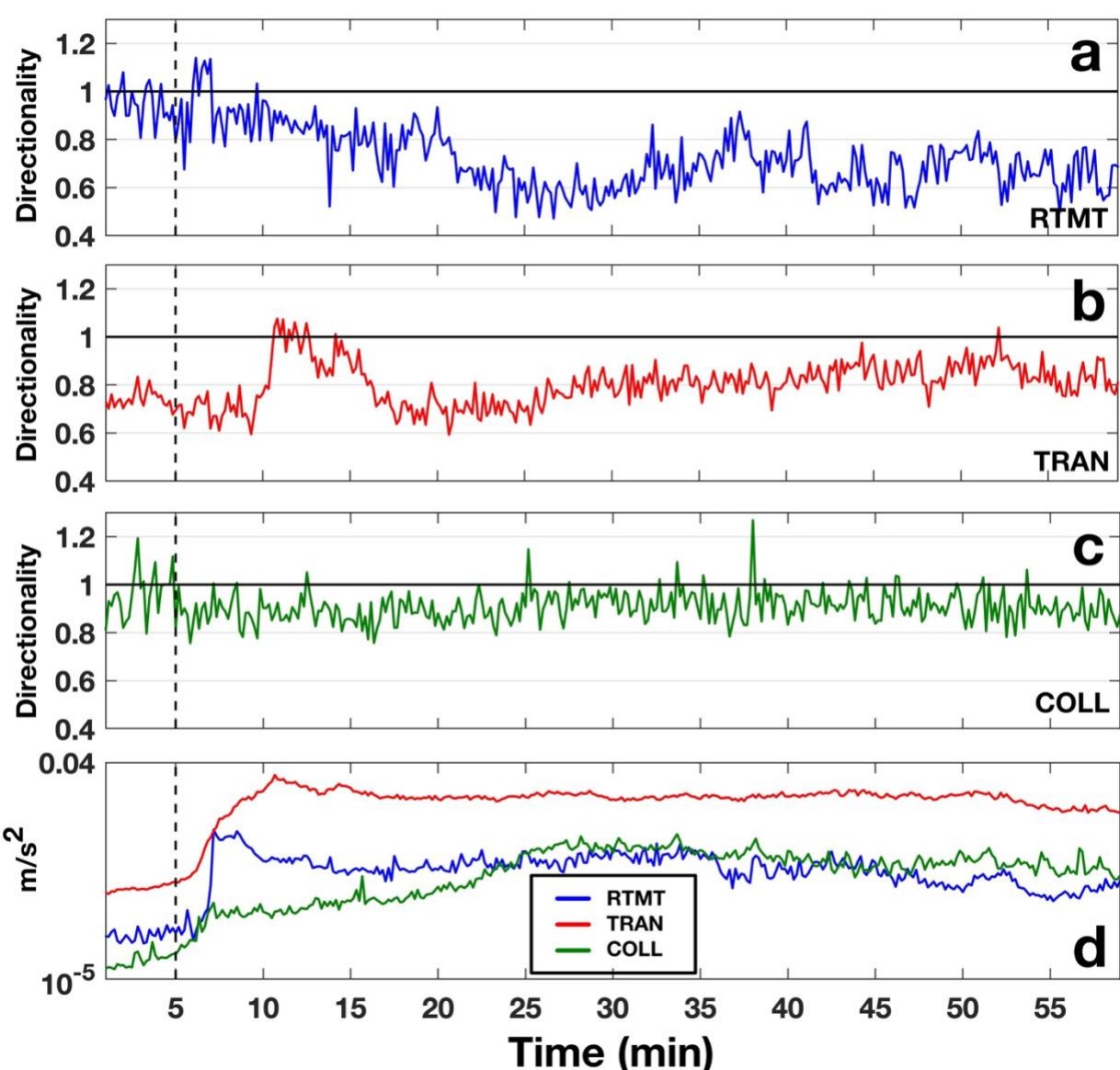


*Figure 6 Directionality ratio plots over time for RTMT (a), TRAN (b), and COLL (c). Vertical RMS seismic signals for*
*the three stations are plotted in (d) where blue is RTMT, red is TRAN and green represents COLL. The dashed*
*vertical lines mark the timing of the lahar front passing the monitoring station. All DRs and RMS amplitudes were*
*calculated using 10 s time windows.*

**278**    **4. Discussion**

**279**    **4.1 Frequency constraints**

**280**    In order to obtain an understanding if PSFs are able to properly describe the lahar dynamics (i.e.

**281**    the weight of the spectral amplitude at the PSF), frequency constraints must be analyzed. To

**282**    complete this, normalized spectrograms along with spectral centroidal frequency (SCF) and

**283**    spectral spreads are computed (e.g. Rubin et al., 2012; Saló et al., 2018). The normalized

**284**    spectrograms are estimated by normalizing (using the maximum) the spectral amplitude for

**285**    each 10 second time window of the lahar individually. By normalizing each time window, ranges

**286**    of dominant frequencies can be visualized. SCFs are used because they represent the weighted

**287**    average of the spectra, and yield the location (i.e. frequency) of the center of the spectral mass.

**288**    The SCF of each time window is estimated similar to that of Saló et al. (2018), in which:

**289**
$$SCF = \frac{\sum_{f1}^{f2} f * A(f)}{\sum_{f1}^{f2} A(f)}$$
(1)

**290**    where $f$ is the frequency and $A(f)$ is the spectral amplitude associated with each frequency bin.

**291**    The spectral spread measures the width of the spectral energy around the SCF (i.e. standard

**292**    deviation), thus yielding information about the quality of the PSFs (e.g. Rubin et al., 2012;

**293**    Giannakopoulos and Pikrakis, 2014; Saló et al., 2018) . Spectral spread can be estimated by:

**294**
$$SS = \sqrt{\frac{\sum_{f1}^{f2} (f - SCF)^2 * A(f)}{\sum_{f1}^{f2} A(f)}}$$
(2)

The computed normalized spectrograms along with SCFs and spectral spreads for each of the
three monitoring stations are shown in Figure 7. For simplicity and comparison, only the flow
parallel data are shown. The normalized spectrograms for every station and component can be
seen in Figures S1-S3, as well as the values in Table S1.
The normalized spectrogram for RTMT (Figure 7a) yields very similar results to that of the PSF
(Figure 3b), where most of the higher spectral amplitudes are at the same frequencies as those
of the PSF. Notably, the low ~10 Hz signal immediately before the arrival of the head of the
lahar is not only seen in the dominant normalized spectra, but also through the decrease in SCF.
Additionally, the PSFs at these time windows are contained within the spectral spread (Figure
7a, black lines). For TRAN, the normalized spectrogram (Figure 7b) is again, very similar to the
PSF in Figure 4b. The SCF mirrors the pattern of the PSF with higher frequencies for the
streamflow, a decrease for the front of the lahar, increase for the head of the lahar, decrease
after the passing of the head, and finally a slight increase later in the lahar body. The
normalized spectra yields this same pattern, with the late lahar body displaying the only
timeframe with increased spectral amplitude distributed throughout the spectral spread (Figure
7b, after 30 min). This most likely explains the bimodal distribution of PSFs for TRAN in Figure 4
after the ~30 min mark. Continuing, the normalized spectrogram for COLL (Figure 7c), also
shows similarities to that of the PSFs in Figure 5b. The PSFs for COLL range between ~20-30 Hz
with a slight bimodal pattern. This same pattern can be seen where the higher spectral
amplitudes are located (Figure 7c). Furthermore, the SCF for COLL splits the PSF range and stays
at ~25 Hz during the bimodal phase of the PSF. Overall, with the analysis of the normalized
spectrograms, SCFs and spectral spreads, we confirm that the use of PSFs to describe mass flow
dynamics is concise for the 18 March 2007 lake-breakout lahar.

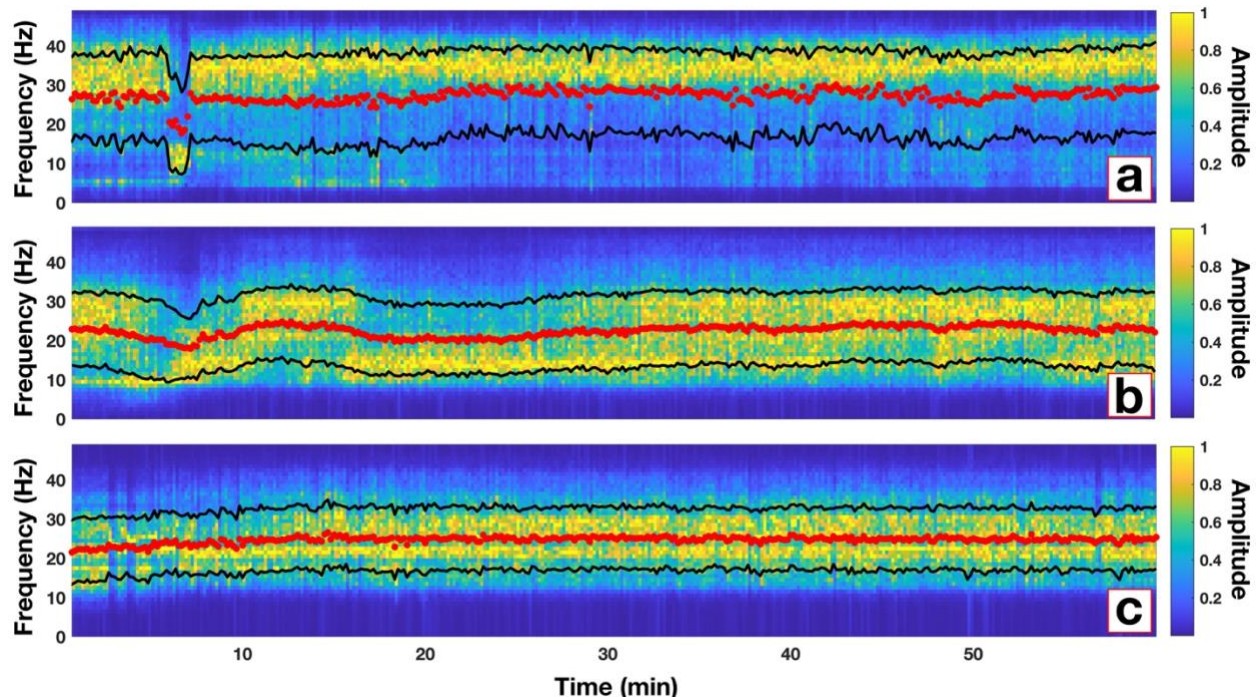


*Figure 7 Normalized spectrograms for the flow parallel direction for each of the three monitoring sites along the*
*Whangaehu channel. Red dots represent the spectral centroidal frequency and black lines show the range of the*
*spectral spread. Note, normalized spectrograms for the other directions can be seen in Figures S1-S3.*
**4.2 Evolution of lahar signals**
*4.2.1 Phase 1 evolution*
A lahar propagating down channel can bulk up by collecting material from erosion or through
the coalescing of multiple pulses to shorten the total length of the lahar (Procter et al., 2010;
Doyle et al., 2011). Lahars can also debulk by depositional means or by the natural elongation
of the lahar as it progresses down channel (Doyle et al., 2011; Lube et al., 2012). Considering
the 18 March 2007 lake-breakout lahar was a large pulse of water that only mixed with the
existing streamflow and contained no juvenile material, examining the seismic signatures along
the flow path can be used to characterize the evolution and transformation of a lake-breakout
event from outburst flood to hyperconcentrated flow and beyond. At RTMT, the seismic
signature is dominated by the flow parallel direction (Figure 3d) with > 30 Hz PSF (Figure 3b).
The exception to this is the timeframe immediately before the head of the lahar passes, when
the PSF decreases to ~10 Hz. This low frequency signal can be seen also at TRAN (Figure 4a-c)
and in the flow parallel direction at COLL (Figure 5b). However, at COLL the PSF is ~20 Hz
instead of 10 Hz as recorded at RTMT and TRAN, most likely due to differing flow properties at
83 km from source. This low PSF before the head of the lahar arrives at each station probably
represents the supercharged stream flow pulse (bow wave, Figure 2c) that is pushed in front of
the head of the lahar (i.e. phase 1, see section 1.1) as described by Cronin et al. (1999) where
they noticed these same pulses in front of lahar heads for three lahars on Mt. Ruapehu in 1995.
Conversely, this frontal pulse could be from the uplift of streamflow from the faster moving
underflow of the lahar (Manville et al., 2000). Furthermore, the low frequency zone before the
head of the flow lengthens as the lahar progresses downstream, suggesting that lahar
elongation can also be seen in the seismic frequency domain (~1 min at RTMT, ~5 min at TRAN).
The ~10 Hz PSF may be explained by flow processes (e.g. frictional resistance of the flow by the
channel, waves at free surface) (Schmandt et al., 2013; Barriere et al., 2015; Bartholomaus et
al., 2015) and could be due to the flow at this stage being more sensitive to discharge (e.g.
increase in shear velocity and/or flow depth) (Gimbert et al., 2014; Schmandt et al., 2017;
Anthony, et al., 2018) or in the case of the underflow hypothesis, frictional sliding on the
channel bed (Huang et al., 2004). The frontal surge or phase 1 of the lahar can be seen in the
DR (Figure 6) as well. For every station along the channel the DR has a slight drop when phase 1
passes the recording station (Figure 6, dashed line). The elongation of phase 1 also has a
correlation with distance from source, where the dip in the DR lasts for only ~1 min at RTMT, ~5
min at TRAN, and approximately 20 min at COLL. The reason the DR decreases during phase 1
for the 2007 lahar could be due to the parallel component being more sensitive to flow
processes than bedload forces (Barriere et al., 2015; Roth et al., 2016). During phase 1
discharge increases, sediment concentration is low (Cronin et al., 1999), and streamflow
dominates resulting in a low DR (e.g. Doyle et al., 2010). The low DR can also be seen before the
arrival of phase 1, due to streamflow already occurring in the channel. The higher flow parallel
amplitude over cross-channel amplitude for streamflow has also been noted in the past for
lahars at Volcán de Colima, Mexico (Walsh et al., 2020).
*4.2.2 Phase 2 evolution*
Following the low PSF phase 1 (i.e. front of the lahar), the peak seismic amplitude occurs (flow
head). The peak seismic amplitude for RTMT (Figure 3d) is accompanied by an increase to
higher PSFs > 30 Hz (Figure 3a-c, 7a). PSFs > 30 Hz have been shown in the past to be either
dominated by turbulence or bedload transport (e.g. Gimbert et al., 2014; Roth et al., 2016). The
2007 lake-breakout lahar has been described as a hyperconcentrated streamflow (e.g. Procter
et al., 2010b) with low sediment concentration, especially early on before the lake water
captured enough material to bulk up and transform. At RTMT, which was only 7.4 km from
source, the lahar had not fully bulked up yet and was in a net depositional regime (Procter et
al., 2010a). Due to the conditions of the lahar at RTMT (e.g. Figure 2a), we surmise the higher
PSF content for the peak seismic amplitude is dominated by turbulent-flow-induced noise.
Furthermore, the higher PSF content at RTMT (>30 Hz) compared to TRAN and COLL (~30 Hz)
could be due to the angle of the slope at the recording stations. Gimbert et al. (2014) noted
that turbulence noise will dominate over bedload-induced noise on steeper slopes due to an
increase in shear velocity. If we use the average flow velocities as a comparison, the lahar at
RTMT (9.3 m/s) flowed faster than at the other two stations (TRAN, 5.6 m/s; COLL, 4.8 m/s).
Further down the channel at TRAN, the PSF for the peak seismic amplitude is ~30 Hz for all
three components (Figure 4a-c, Table S1). Again, this high PSF may be attributed to turbulence,
as seen by the images taken at TRAN (Figure 2d). The difference at TRAN is the duration of the
higher PSF, where at RTMT the high PSF stays throughout the entirety of the recording window,
at TRAN the high PSF only last for ~5 min (Figure 4a-c, ~11-16 min). The difference at TRAN
could be from the evolution of the lahar. By time the lahar reached the monitoring station at
TRAN (28 km from source) the lahar was fully bulked up and had the properties of a traditional
four phase lahar as described by Scott (1988) or Cronin et al. (1999) (Figure 2c-e, see section
4.2.3). By time the lahar reached COLL 82 km from source (Figure 5), the peak seismic
amplitude is associated with PSFs between 15-30 Hz, with bimodal patterns in the horizontal
components and a tighter spread in the vertical component (~27-29 Hz). At COLL, the lahar had
converted into a plug-like flow with lower turbulence and hence the higher PSFs are most likely
associated with bedload transport (Figure 2f). Furthermore, Burtin et al. (2010) and Roth et al.
(2016) noted that when the vertical component has greater seismic amplitudes than the
horizontal components, bedload dominates. This same amplitude feature can be seen at COLL
(Figure 5d, past ~25 min) where the vertical energy is greater than each of the horizontal
components. The bimodal frequency pattern of the horizontal components (Figure 5a,b) is
likely to be the recording of both water-flow noise (lower PSF) and bedload transport (higher
PSF). This also explains why the vertical component does not show the same bimodal frequency
pattern. Barriere et al. (2015) described the parallel component as being more sensitive to flow
properties (e.g. discharge, depth, shear velocity), and Doyle et al. (2010) noted that the cross-
channel component is likely dominated by the amount of turbulence (water and particles acting
on the channel walls), thus the reasoning behind the differing PSF patterns between
components. This PSF feature is similar to the lahars recorded by Walsh et al. (2020), where the
cross-channel PSF is confined within a narrow band around 15-20 Hz and the flow parallel PSF is
more bimodal (10-40 Hz). At COLL, the cross-channel PSF (Figure 5a) is dominated by PSFs at
~18 Hz (lower than vertical component at ~28 Hz, Figure 5c), with the flow parallel between 20-
30 Hz (Figure 5b).
The DR at the peak seismic amplitude for all three recording stations increases (Figure 6). The
DR for both RTMT and TRAN increases to DR > 1. Doyle et al. (2010) noted that higher wetted
perimeters will increase the DR, which can be seen at TRAN for the 18 March 2007 lake-
breakout lahar (Figures 2d, 6 peak DR/RMS amplitude). Conversely, the DR decreases after the
peak seismic amplitude while the wetted perimeter is still high (Figure 2d,e). While the wetted
perimeter may be a factor in increasing cross-channel energy and thus the DR, the more likely
explanation for the 18 March 2007 lahar might be the higher level of particle collisions and
turbulence at the peak seismic amplitude. More turbulent particle collisions would increase the
DR (e.g. Doyle et al., 2010) due to more lateral excitation within the flow and against the
channel walls increasing the cross-channel signal. The increase in collisional energy also relates
well with the PSF, as higher PSF correlates to an increase in the amount of interflow collisions
as shown by Huang et al. (2004), and may also explain the slight increase in DR overall when the
PSF increases (Figure 8a). The DR for COLL (Figure 6c) during this same timeframe probably is
not due to the amount of particle collisions due to the plug-like flow (Figure 2f), but rather the
increase in sediment concentration (Figure 8c). As the sediment concentration increases at
COLL the DR starts to increase as well (Figure 8b). Similar to Doyle et al. (2010), COLL yields a
correlation between DR and sediment concentration ($R^2$=0.95, Figure 8b), where higher DRs
indicate higher concentrations of sediment contained in the flow. Lastly, as noted above, DRs
may correlate with PSF or at least indicate differing processes taking place within the flow
(Figure 8a). Lower PSF would produce lower DRs because low PSF are more sensitive to water-
flow processes (hence higher parallel energy), whereas higher PSFs would produce higher DRs
due to higher PSF being dominated by sediment, particle collisions and turbulence (higher
cross-channel energy) (Figure 8a).

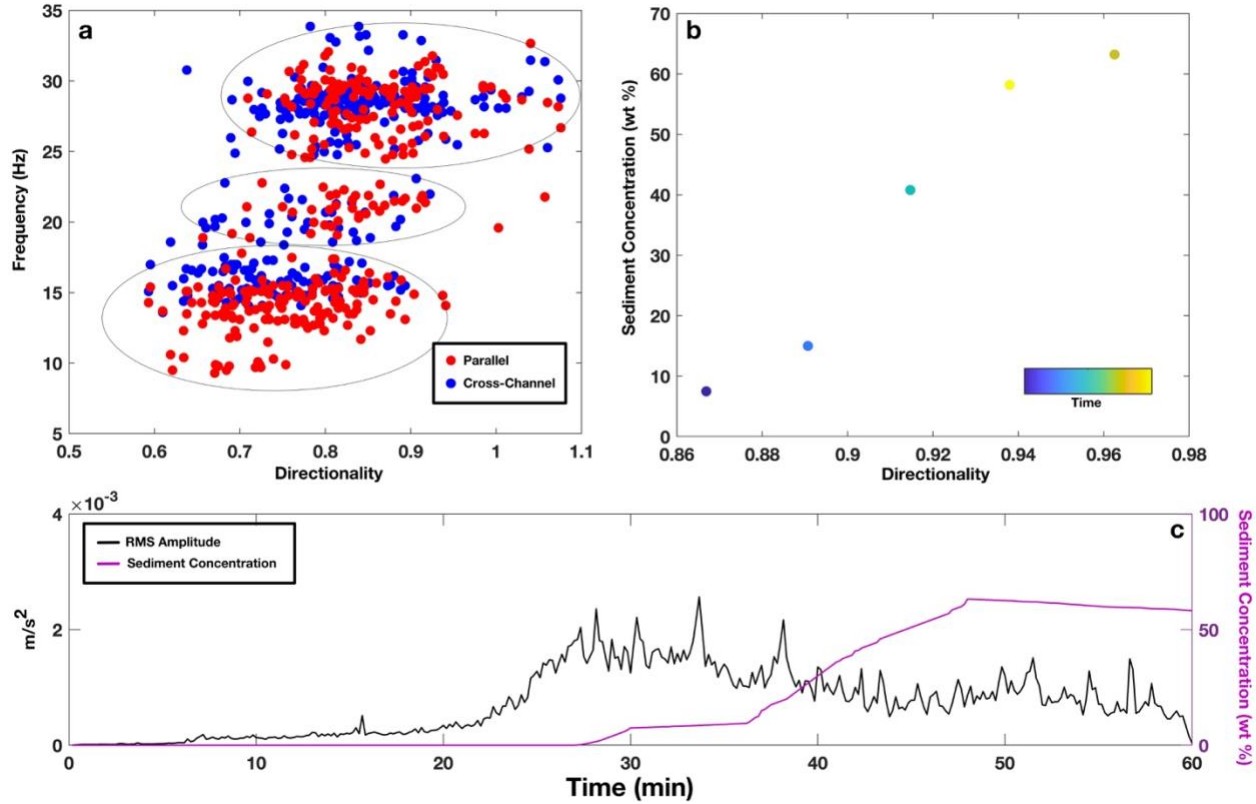


*Figure 8 Plots of (a) comparing PSF and DR at TRAN, (b) sediment concentration and DR at COLL ($R^2$=0.95), and (c) seismic amplitude (black line) with sediment concentration (purple line) depicting the lag in sediment at COLL. Note on (a) parallel (red dots) and cross-channel (blue dots) PSF display three different zones (black circles). Also note that at COLL the first sediment concentration measurement did not occur until the 30 min mark.*

*4.2.3 Development of flow phases at TRAN*

While the lahar at RTMT was a large outburst flood/sediment-laden flow, and at COLL a plug-

like flow, at TRAN the 18 March 2007 lahar was a dynamic bulked up lahar (see Figure 2b-e).

The evidence for this is in the PSF content for TRAN (Figure 4a-c) compared to the other two

monitoring sites (Table S1). At TRAN the PSF has a step-up step-down pattern for the first 30

min of the lahar passing, and then transitions to a bimodal or wide PSF range for the rest of the

recording window. As noted above, the low PSF preceding the lahar head arrival is thought to

be due to a sensitivity to water transport properties (Figure 2c, phase 1). The increase to higher

PSFs during the peak seismic amplitude may be from particle collisions and/or higher

turbulence (Figure 2d, transition from phase 1 to 2). After the maximum seismic amplitude at
TRAN (Figure 4, ~10-15 min), the PSF decreases to 10-20 Hz. This drop in PSF after the highest
stage and amplitude could be from a more water-flow dominate regime (seen in the increased
parallel amplitude, Figure 4d, and decrease in DR, Figure 6b), where turbulence decreases
(Figure 2e), discharge is still high, and the peak sediment concentration has not occurred yet
(e.g. Cronin et al., 1999). Likewise, the decrease may also be from greater frictional sliding on
the channel bed (Huang et al., 2004). After the decrease to 10-20 Hz PSFs, the PSF displays a
bimodal or wide frequency range at ~28 min (Figure 4a-c, 7b). As aforementioned for COLL, this
PSF pattern could be from both bedload- and water-flow-induced noise. This timeframe (phase
3) is also where the peak sediment concentration would be (not recorded at TRAN), as noted by
Cronin et al. (1999), and thus the PSF would show more bedload high PSF. This hypothesis also
compares well with the DR (Figure 6b), where the cross-channel energy increases starting at
~25 min indicating that the sediment concentration may be increasing (Doyle et al., 2010).
Finally, the wide PSF range later in the recording window (Figure 4) could also result from the
lahar having two distinct layers as described by Cronin et al. (2000), where there is a wide more
dilute finer grain top layer and a channelized sediment-rich layer on the bottom. The two layer
model can apply to TRAN because the lahar at this monitoring station overtook the channel
(Figure 2d,e) and proceeded to flow horizontally outward forming the surface layer described
by Cronin et al. (2000).
**4.3 Implications for monitoring**
The main goal of this research is to contribute in defining better monitoring criteria for
dangerous mass flow events. The data described above is part of a larger collection of
monitoring data collected over the entire length of the Whangaehu channel consisting of 21
monitoring sites and years of preparation (e.g. Manville and Cronin, 2007; Keys and Green,
2008). Due to this, the ability to accurately estimate the properties of the lahar at various
stages along its path is possible. When it comes to flow events of any size, the ability to
understand how the dynamics change with distance along the channel is important for warning
and future hazard mitigation. We show here that a lake-breakout event can start out as an
outburst flood, bulk up into a hyperconcentrated flow, then eventually elongate and entrap
enough sediment to transform into a plug-like slurry flow. Each of these flow types yields
differing PSF ranges and patterns due to the relationship between the channel geometry,
sediment concentration, turbulence, and bedload transport. While the lahar at different
stations along the channel may have differing PSF content, we also show that the lahar
elongates and a predictable model (e.g. Cronin et al., 1999) can be used with and shown in the
seismic data. Being able to apply such a model may yield some relevance of universality in
terms of warning systems at different distances away from the mass flow source. Whereas,
shown above, the flow phases at each monitoring station can be seen, but at differing lengths
and times in the seismic signal (e.g. Figure 6). To better visualize this concept, conceptual
models based off of the Cronin et al. (1999) models are created for each of the three seismic
stations for the 18 March 2007 lahar (Figure 9). In the conceptual models for the 2007 lahar,
the aforementioned elongation of the frontal pulse or bow wave (phase 1) and head of the
lahar (phase 2) is shown, along with the differences and similarities between the properties of
the lahar at the three seismic monitoring sites.

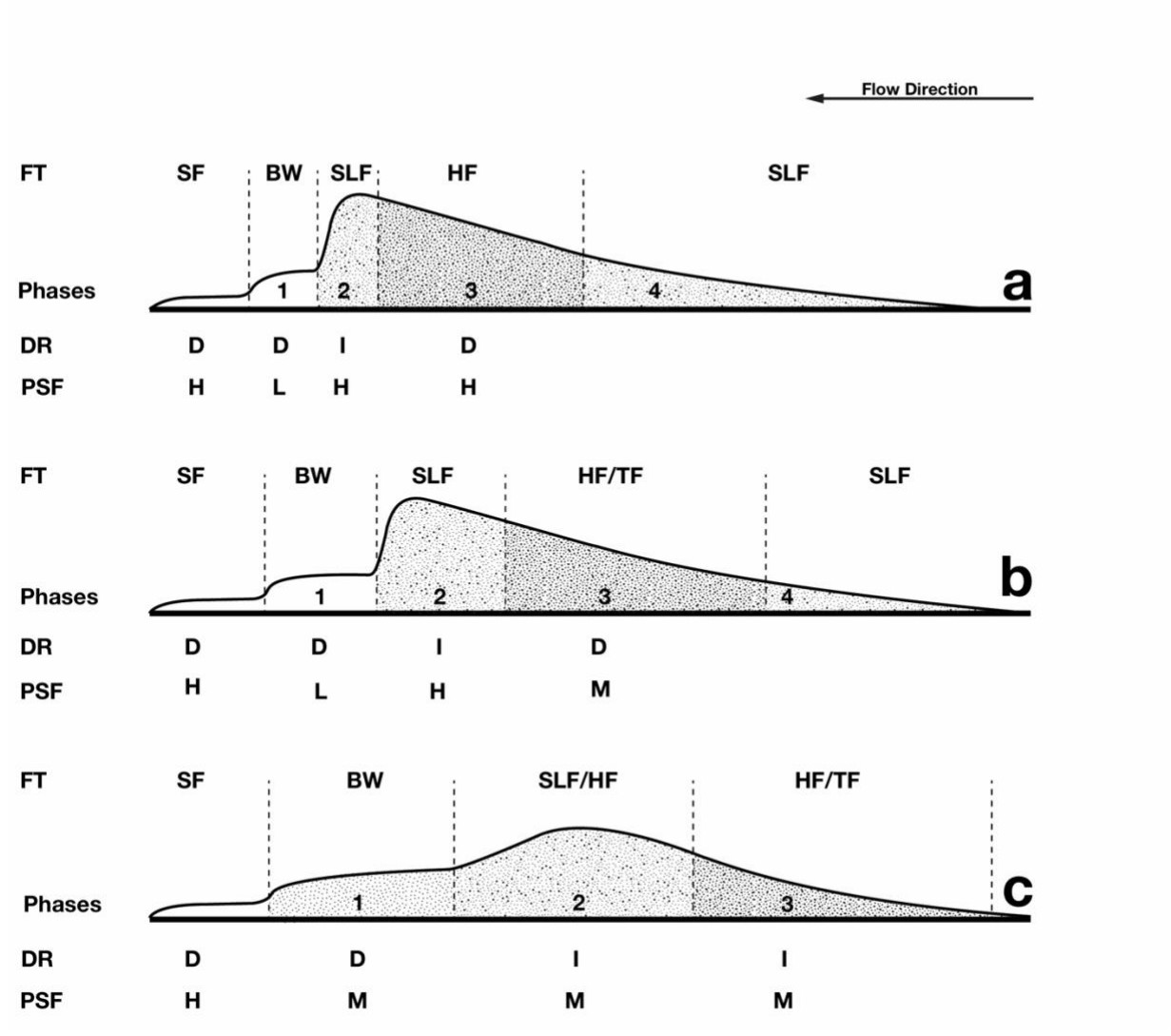

*Figure 9 Conceptual models for the 18 March 2007 lahar at each of the three monitoring stations along the*
*Whangaehu channel depicting flow type and the estimated seismic properties at each flow phase. a) RTMT 7.4 km*
*from source, b) TRAN 28 km from source, and c) COLL 83 km from source. Flow types (FT) are as followed;*
*streamflow (SF), bow wave streamflow (BW), hyperconcentrated flow (HF), Transitional flow (TF), and sediment-*
*laden streamflow (SLF). Note, decreased (D), increased (I), high (H), low (L), and mixed (M) are notations for*
*directionality ratios and peak spectral frequency estimates. See Table S1 for value ranges for each property.*
Another implication for future warning is the implementation of 3-component sensors and the
use of DRs for channels that have streamflow. Walsh et al. (2020) showed for lahars flowing in
La Lumbre channel at Volcán de Colima that the DR for streamflow is <1 and then increases
when the head of the lahar arrives. This same feature can be seen at each of the three
monitoring sites for the 18 March 2007 event (Figure 6) indicating differing flow types will still
show this DR pattern within the same flow and at other channels. To further show this, there
were three natural non-lake-breakout eruption-based lahars that occurred in the Whangaehu
channel in September 2007 (for more details on the lahars see Cole et al., 2009; Kilgour et al.,
2010) and recorded on the seismometer at RTMT. The DR for the September events starts with
streamflow with a DR < 1 and when the first lahar arrives the DR increases to >1 and as the
lahar fully passes, the DR decreases to <1 again (Figure 10a). As the second lahar arrives at
RTMT (Figure 10, second dashed line), the DR increases to >1 again. After the second lahar
passes the DR deceases once again back below DR<1. Finally, as the third lahar arrives (Figure
10, third dashed line) the DR yet again increases above 1 for the entirety of the event.
For many mass flows and especially those that flow into channels with preexisting streamflow,
the peak seismic amplitude does not always coincide with the arrival of the mass flow, and thus
may not be the most reliable for event detection or warning (e.g. Arratano and Moia, 1999;
Cole et al., 2009). These observations may be due to a frontal surge, the lag in sediment
concentration or differences in peak amplitude with peak discharge. Phase 1 (frontal
streamflow surge) of the model proposed by Cronin et al. (1999) was based on a
hyperconcentrated flow interacting with streamflow, but has also been shown for debris flows
as well (e.g. Arratano and Moia, 1999). Arratano and Moia (1999) showed at Moscardo Torrent,
Italy, through a hydrograph that there was a precursory surge ahead of the debris flow that was
not seen in the seismic record. Similarly, at Ruapehu, for the 18 March 2007 lahar, at each of
the three stations there is little evidence or rise in the seismic amplitude that would indicate
that there was a precursory surge or phase 1 (Figures 3-5, bottom panel), which could be
problematic for detection methods that use amplitude thresholds or short-time-average vs
long-time-average (STA/LTA) algorithms. Conversely, the surge ahead of the lahar can be seen
in both the PSF analysis (drop to low frequencies) and in the DR (decrease in DR) right before
the peak seismic amplitude arrives. This shows that when monitoring for future events that not
only the amplitude should be used, but other analysis (e.g. PSF, DR) as well, otherwise there
could be a delay in the detection of an event.

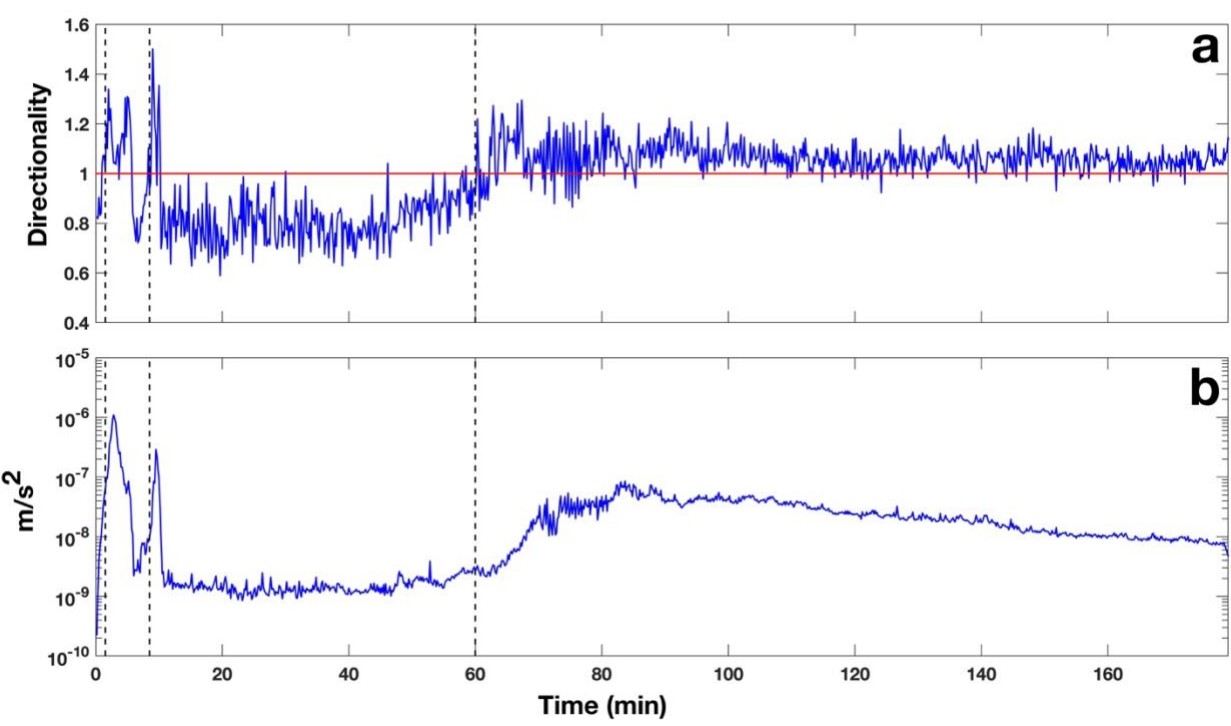


*Figure 10 (a) Directionality ratio for the time sequence of the three lahars that occurred on 25 September 2007. (b)*
*RMS amplitude of the seismic record at RTMT during the timing of the three September lahars. Note that the black*
*dashed lines represent the timing of each lahar arriving at the monitoring site.*
Using all three components of the seismometer can be very beneficial in lahar monitoring. The
above-mentioned DR analysis can only be completed with horizontal recording, and analyzing
PSF in each component can yield critical information about the flow properties and dynamics.
Examining the seismic amplitude differences can generate significant discoveries, for example,
when the vertical component is stronger than the horizontal components, bedload may
dominate over turbulence noise (Burtin et al., 2010). Greater flow parallel signals may indicate
higher water transport noises (Barrier et al., 2015), while higher cross-channel signals could be
caused by increased interflow particle collisions and flow-channel wall interactions (Doyle et al.,
2010). While using the differences in each component can be useful, there are also some
concerns. Channel geometry and bed conditions can alter the seismic signal (e.g. Coviello et al.,
2019; Marchetti et al., 2019). Additionally, the flow parallel direction can be influenced by the
lahar that has already passed, the lahar at the station and the lahar arriving. Furthermore, the
tilt of the seismometer may play a large role in determining which component is stronger (e.g.
Anthony et al., 2018). In the case of the 18 March 2007 lahar a large pulse of water passed the
monitoring stations which may explain why the parallel component is stronger than the other
two components at RTMT (Figure 3d) and TRAN (Figure 4d). At COLL, the lahar had elongated,
lost energy, and thus shows more decreased flow parallel energy compared to the previous two
stations (Figure 5d). In the cross-channel direction, if a flow overtops the channel, the
amplitude would presumably be dampened. This may be the case at TRAN where both the flow
parallel and vertical directions are more energetic than the cross-channel amplitude after the
passing of the head and breaking out of the channel occurred (Figures 2d, 4d). Another concern
when using the horizontal components of a seismometer are the effects shallow layers may
have on the site response of the sensor. This is especially true when a sensor is installed on soft
or lose sediment (e.g. soil, fluvial/alluvial deposits). To test for potential effects by shallow layer
fundamental frequencies, H/V analysis of ambient noise (streamflow dominant) was conducted
(see Supplementary material). For RTMT, the H/V results depict a broad frequency peak
between 5-15 Hz with a local maximum at ~8 Hz (Figure S4a). Comparing the H/V frequency
with the PSF of RTMT (Figure 3), the only overlap is when the front of the lahar passes the
station where the PSF decreases for ~1 minute before the head of the lahar arrives. The H/V
analysis for TRAN has a multi-broad-peak shape, with frequency peaks at ~14 and ~28-35 Hz
(Figure S4b). While these frequencies are similar to PSF values for TRAN (Figure 4), the H/V
analysis has no distinguishable fundamental frequency, contains large error, and no frequency
peak has a H/V amplification > 2. In order for a H/V frequency peak to be considered ideal,
generally the amplification must be greater than 2 and the standard deviation lower than a
factor of 2 (SESAME, 2004). The H/V amplification for COLL displays a broad frequency peak
between 13-18 Hz, with a local maximum at ~18 Hz (Figure S4c). Comparing the PSFs at COLL
(Figure 5), only the cross-channel direction has significant PSF values in the same frequency
range (~18 Hz band). With all three stations not yielding distinct H/V fundamental frequencies,
we surmise that the PSF content for the 18 March 2007 lake-breakout lahar is most likely
dominated by the large flow passing by the seismic sensor rather than large site amplification
effects from a shallow layer. While this may be the case, there is still the possibility that some
of the PSF values could be due to local effects and should not be considered in the lahar
analysis, e.g., the low PSFs at RTMT between 15-20 min (Figure 3a,b), at TRAN contributing to
some of the "jumping" in PSF content (Figure 4a-c), or in the mostly dominant 15-20 Hz PSF in
the cross-channel direction at COLL (Figure 5a). Conversely, SCF values at each station do not
reside in the broad H/V frequency range at any station (Figure 7), which may further support
the hypothesis that almost all of the recorded frequencies are indeed produced by the lahar.
With the use of horizontal components becoming common in mass flow monitoring, future 3-
component analyses of mass flows should consider estimating H/V ratios or use other site
response methods (e.g. spectral ratio analysis) in order to identify whether near-surface
structures may affect the recorded flow data. Overall, all these concerns can and should be
tested to estimate potential error in 3-component methods. Nevertheless, using all three
components of the seismometer can enhance the productivity of warning systems, and if
possible, should be used instead of single component sensors.
Finally, implementation of these new results into new or existing mass flow warning systems
must be discussed. In an ideal setup, to remove any doubt about the recorded signal, machine
learning techniques should be used to separate the mass flow noise from other non-flow noises
(e.g. environmental, human induced, earthquakes). For instance, recently Wenner et al. (2021)
used a supervised random forest algorithm to classify differing sources in a debris flow setting.
Once the mass flow source has been classified, integrating automated DR and PSF analysis
would be quick and straightforward. Implementation of these techniques would be similar to
other seismic analysis or detection methods, such as a STA/LTA (e.g. Coviello et al., 2019) or a
number of frequency detection algorithms (e.g. Rubin et al., 2012) where real-time analysis of
set time windows are used to determine if there has been a change in the seismicity along the
channel. The system could be programed to identify changing features in the flow automatically
by analyzing the content of each window, as well as comparing previous time windows. The
analysis of continual data could then be feed into machine learning algorithms (e.g. Rubin et al.,
2012; Wenner et al., 2021) to increase the confidence of not only detection, but the
characterization of flow behavior. One of the main discoveries of this research was the
evolution of seismic signals produced by the lahar as the flow moved further from its source.
The changes in the seismic signal along with the flow characteristics may be able to help hazard
and forecast modeling through the use of numerical models (e.g. Mead et al., 2021). Modern
flow hazard assessment is based on numerical models that use potential energy equations of a
large non-changing mass sliding down slope with limited inputs for how the flow may evolve
over time as starting inputs. This can lead to errors in risk mitigation and hazard assessments.
The findings shown above on how the 18 March 2007 lahar evolved over 83 km at Mt. Ruapehu
will help to improve mass flow modeling in the future by enabling modelers to add constrains
or more inputs on how a mass flow might evolve, leading to improved forecasts and hazard
assessment.
**5. Conclusions**
At 23:18 UTC on 18 March 2007, Mt. Ruapehu produced the biggest lahar in New Zealand in
over 100 years causing $1.3 \times 10^6$ m$^3$ of water to flow out of the Crater Lake and rush down the
Whangaehu channel flowing for over 200 km to the Tasman sea. Seismic analysis at three
monitoring locations along the channel (7.4, 28, and 83 km) yielded an understanding of how
flow type and processes of the lahar evolve with distance. The proximal lahar was a highly
turbulent outburst flood, which generated high PSF content in all three components. Further
along the channel after the lahar had bulked up and transformed into a multi-phase
hyperconcentrated flow, the PSF content was variable and showed changes in the flow
regime/phase. Finally, at the most distal monitoring station, the lahar had lost energy and
transformed into a slurry-type flow where the PSF content became more bedload-dominant.
Additionally, directionality ratios from all three sites along with data from additional lahars
yielded strong evidence that DRs can be used for warning systems when there is streamflow
present in the channel. Furthermore, PSF and DRs show evidence of a pre-lahar water pulse
that may be concealed in the raw seismic data, but has been observed visually. Ultimately, the
use of 3-component broadband seismic analysis for the 18 March 2007 lahar at Mt. Ruapehu
may lead to more accurate and advanced real-time warning systems for mass flows through the
use of frequency and directionality around the world.
*Author Contribution*
BW performed seismic analysis and drafted the manuscript, CL organized and prepared data,
and JP created the visual location representation of the event. All participating authors
contributed to the discussions and editing of the draft of the manuscript, as well as approving
the final edition.
*Competing Interests*
The authors declare that they have no conflict of interest
*Data Availability*
The data used in this publication can be found at: Braden Walsh; Charline Lormand; Jon Procter;
Glyn Williams-Jones (2022), "18 March 2007 Mt. Ruapehu lahar seismic data,"
https://theghub.org/resources/4890.
*Acknowledgements*
This work was supported by the Resilience to Natures Challenges – New Zealand National
Science, volcano program of research. We would also like to thank all the people from Massey
University, Horizons Regional Council, NIWA, and the Department of Conservation that
collected data and set up monitoring locations all along the channel in preparation for and
during the lahar. A final special thanks to Kate Arentsen for editorial support.

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
