# Peer review of "Characterizing the evolution of mass flow properties and dynamics through analysis of"

_Natural Hazards and Earth System Sciences, 2022_

## Author Response (AR1)

**Addressing Reviewer Comments:** Reviewer in **bold** text and response in regular text.

Reviewer 1:

**Major comments**

**At that point I confirm myself that to draw valid conclusions the authors have to take into account the amplitude of the frequency peaks At that point I confirm myself that to draw valid conclusions the authors have to take into account the amplitude of the frequency peaks**

While the amplitude of each frequency range of the signal will yield differing content and the contribution amount will be different, the results and especially the discussion section of this manuscript are still valid. The use of PSF or dominant frequencies has been used by numerous studies on a variety of mass flows (of which many are cited in the manuscript) to estimate the dynamics. The "contribution" of the PSF or difference between the spectral centroid yields the same results and is also an indicator of dynamics based on frequency (i.e. when PSF is bimodal/"spread" is large (contribution small), multiple processes are occurring). That said we added a section in the discussion looking at the normalized spectrograms as well as the centroidal frequency to show when and where the PSF is most dominant.

**Authors must include data processing section/subsection as well as an explanation of how the data results are obtained.**

All of this information is already in the manuscript. Most is in the "Data" section as well as Lines 164-169 in the "Results" section. As for equations, we believe it is not necessary to include basic equations for converting to frequency domain or directionality (simple division). For the new SCF analysis we included equations, see new section 4.1.

**To facilitate the interpretation of the results the authors must include a table with the values and characteristics observed in the figures as well as indicate in the figures the different parts described in them.**

We describe and include multiple images from the lahar in figure 2 as well as created a model in figure 8 referencing the seismic data with the observations. Furthermore, throughout the discussion we reference each figure multiple times while explaining the results. See lines 367-377, 386-388 for example.

**Line by line comments**

**Line 68: seismic instruments have been used since 1998, I do not consider it relatively young.**
Changed to "However, in order to fully utilize these instruments, improved interpretation, comprehension, assessment, and universality is needed."

**Line 69: specify. There are more geophysical instruments, you are only referring to those relating to vibrations.** See change for Line 68

**Line 83: you must include the purpose of this contribution because the previous ones also deal with this topic.** The inclusion of section 1.1 is to give the audience an introduction on the details of the parts of the conceptual lahar model on Mt. Ruapehu. Which then transitions into section 1.2 which talks about the 2007 event. The first paragraph in the manuscript is a brief introduction on mass flows and why it is important to monitor them.

**Line 130: Homogenize or clarify the terms channel, river according to the figure also throughout text.** We homogenized the wording throughout.

**Line 144: change the sentence.** Changed to "supplementary measurements"

**Line 155: in the instrument setup is the vertical component of the instrument orthogonal to the slope or zenith direction. In our studies we have observed that the results are different. Indicate, at least the angle of the slope where the sensors are installed.** Added "Furthermore, the seismometers were installed normal to horizontal to lessen the degree of vertical energy transfer to the horizontal components."

**Line 162: Are other instruments co-located with the seismometer? How are the average speeds determined. Also explain the arrival times of the lahar.** Added "Arrival times are based off of images and eye witnesses at each of the monitoring stations. The flow velocity at RTMT and COLL were estimated from imagery and at TRAN from a flow meter."

**Line 193: in some cases, the event passes over the station, this is not the case. The lahar passes closer to the station, in fact it is the record at the station of the waves generated by the passage of the lahar.** Deleted "passing the station"

**Line 236: Are you considering amplitude of the signal or the energy.** Changed to "amplitude"

**Line 250: note that it is shorter in RTMT than in TRAN.** Noted in text

**Line 251: notice that there is a bend in both curves of the same wavelength but shifted in time. Note the similar concave behavior at RTMT and TRAN.** Further analysis has shown no correlation with this "curve" in DR at RTMT and TRAN.

**Line 254: Describe properly minute 7.** Minute seven is described on lines 252-253

**Line 568: fix citation for Lube et al., 2012:** Fixed citation

Reviewer 2

**Major comments:**

**Site amplification:**
The time windows prior to the lahar arrival has similar PSF due to the fact that there was already streamflow in the channel and it has been shown in the past that streamflow has higher PSFs. The March lahar was sourced from a lake breakout and hence would show similar PSF to streamflow compared to a more traditional bulky debris flow. To your concern with site amplification, we went ahead and estimated H/V ratios for each station. This analysis of where the H/V frequency peaks are has been included in the revised manuscript.

**Frequency analysis:**
See response to first major comment from Reviewer 1. We calculated centroidal frequencies and normalized spectrograms to compare to the PSF results. More details on the actual data processing were added as well.

**Physical reasons for signal properties**
We cite many papers in which data came from natural and laboratory sources that describe the physical reasons why the seismic signals change. On this point, we describe these throughout the manuscript many times and how they relate to our signals. We have gone back through the manuscript and rewrote some areas to better clarify these statements.

**Line by line comments:**
**Line 61-63: a lot of these things are true of seismic instruments as well, and there is more ambiguity in interpretation for quantitative values. I also disagree that they can be used for "accurate" (L66) estimates of flow properties. Only in very limited situations is that true.**
Changed to "and/or lack the capability to evaluate multiple pulses or flow events"

**Line 69-70: Using seismometers for flow monitoring is not young:** Changed to "However, in order to fully utilize these instruments, improved interpretation, comprehension, assessment, and universality is needed."

**Line 77: previously not recorded by who? People have used three components many times in the past:** changed to "yield additional information about the flow that is not utilized if only the vertical component is used"

**Line 88-89: Perhaps it would be useful to explain what you mean by terms like plug-like and laminar:** These are explained in the references, as well when describing the lahar at COLL

**L133: missing "the" and missing comma.** Added

**L133-134: please explain how velocities are measured**. Added how the velocities were estimated. See replies to comment from Reviewer 1.

**Line 157: explain what the averaging represents.** Added "The flow velocity at RTMT and COLL were estimated from imagery and at TRAN from a flow meter."

**Line 166-167: give details as to how the PSF was estimated.** Figures 3-5 show the peak spectral amplitude at its represented frequency. Line 166-167 describe how the points were estimated. To add value to this please see the new supplementary figures in which normalized spectrograms for each station and component are displayed.

**Line 169: please explain how the arrival time is known.** Added "Arrival times are based off of images and eye witnesses at each of the monitoring stations." On line 162.

**Line 180-181: how do you know this is the arrival of the head? Why is the word streamflow in parentheses?** Line 180 states "prior to the arrival of the head (peak seismic amplitude)." The streamflow was meant to state streamflow is in the channel. We see the confusion. Deleted "(streamflow)"

**Figure 4: it's interesting that there is an upward sweep of frequency on the vertical component, any idea why?** There is a sweep in all the components, but the vertical shows the most consistent, probably due to the better coupling with the ground compared to the horizontal directions and what controls them.

**Line 232: rephrase this sentence:** changed to "so that the North component is aligned to be parallel to the flow".

**Line 234: site effects cannot be ignored.** See comment to major comment.

**Line 236: give details on how the energy was computed and directionality**. Changed to "The directionality ratio (DR) can be defined as the cross-channel amplitude divided by the flow parallel amplitude."

**Line 237-241: add information here about physical reasons why the directionality would contain information about rheology changes.** Added information about differences in signal between streamflow and lahar.

**Line 265-266: I don't understand what multiple pulses has to do with bulking material that is differing from collecting material from erosion, maybe rephrase sentence.** Changed to "or through the coalescing of multiple pulses to shorten the total length of the lahar"

**Line 284: this is a vague statement, can you be more specific?** The explanation is stated following this statement on lines 285-288.

**Line 305: can you explain what is meant by a 4-phase lahar.** Added "(see section 1.1)"

**Line 364-365: here and elsewhere, are the speculations about flow style corroborated by the camera images or other data types that were collected**. See figure 2 and refences to figure 2 throughout the manuscript. Also figure 7 for data relating to COLL. There were also eyewitnesses to confirm the camera images.

**Line 405: unclear what is meant by "at different distance away from source".** Source is the starting point of the flow event. Changed to "the mass flow source"

**Line 435: This statement and the supporting evidence is one worth emphasizing more in the paper.** We added some lines throughout to emprise this.

**Line 444-446: is the change in directionality unique to when a lahar is passing by? Could it be differentiated from other seismic sources?** This is an interesting idea and needs to be looked into in the future, but is outside the realm of this study, but would be a great future study to do. That said, the channel in question in this manuscript always contains streamflow, so the DR is always at "streamflow" levels when only recording "background" noise. Furthermore, DR of background noise in a "dry" channel was recorded at Colima, Mexico and was always high and indistinguishable from lahar DR, see Walsh et al., 2020.

**Line 464-466: tilt is usually at much lower frequencies. Since no details were given on how the energy was computed for each component it is hard to assess whether this would have an influence or not**. Details are given in the Data and Results sections. Also see the citation given that lead to the statement of tilt. Tilt has also been used as a detection method for mass flows in the past as well.

---

## Referee Report (RR1)

This work focuses on seismic analysis at three monitoring locations along the channel yielding an understanding of how flow type and processes of the lahar evolve with distance. I found this article very interesting and technically correct. The problem is sufficiently described and motivated and the analysis of the results perfectly documented. Perhaps the only drawback, in my opinion ,is its applicability in real-time and risk situations.

Both the analysis and the conclusions are based on the processing of seismic signals before and after the appearance of the event itself. That is why my only doubt is based on how this work could be included as a real-time monitoring tool that improves or accompanies video-cameras. The study of seismic signals offers very important information on the phases of the flow as well as how it evolves with distance, but these conclusions are obtained from 'medium-term' analysis.

How should this technique be implemented in an observatory and how should it be analyzed to obtain relevant conclusions in the short term? Could this technique be automated in order to extract conclusions about the flow without human supervision?

I think these questions should be addressed in the manuscript in order to improve itself.

Finally,I have also found some minor bugs and some suggestions that will help to improve the work.

1) Section 3.1, line 187, Figure 2 is referenced. I think this reference is a mistake, and the authors wanted to reference Figure 3.
2) Section 4.2, line 325, the authors say: ' This low PSF before the head of the lahar arrives at the station could represent the supercharged stream flow pulse..'. What station are you referring to, RTMT? Whilst COLL station is being analyzed the analysis of the RTMT station is addressed, which could lead to confusion. It would be interesting to review this paragraph.
3) Finally, also in section 4.2, line 340, when the DR in COLL is analyzed, the authors refer to a phase change around 20 min after the arrival of the flow. Honestly, I am not able to visualize such a change in Figure 6. A similar change occurs around minute 12. Could the authors explain this conclusion?

---

## Author Response (AR2)

**Addressing Reviewer Comments:** Reviewer in **bold** text and response in regular text.

Reviewer 3:

**Comments**

**Both the analysis and the conclusions are based on the processing of seismic signals before and after the appearance of the event itself. That is why my only doubt is based on how this work could be included as a real time monitoring tool that improves or accompanies video cameras. The study of seismic signals offers very important information of the phases of the flow as well as how it evolves with distance, but these conclusions are obtained from 'medium-term' analysis. How should this technique be implemented in an observatory and how should it be analyzed to obtain relevant conclusions in the short term? Could this technique be automated in order to extract conclusions about the flow without human supervision?**

We have added a paragraph at the end of the discussion in section "Implication for monitoring". We discuss how DR and PSF could be integrated into monitoring systems and how knowing the evolution of the flow signals can help hazard mitigation.

**Line by line comments**

**Line 187: Figure 2 is referenced, I think this reference is a mistake, and the authors wanted to reference Figure 3**. Changed to "Figure 3"

**Line 325: The authors say "This low PSF before the head of the lahar arrives at the station could represent the supercharged streamflow pulse…" What station are you referring to RTMT? Whist COLL station is being analyzed the analysis of the RTMT station is addressed, which could lead to confusion. It would be interesting to review this paragraph.** We see the confusion the wording may have had, we changed the sentence to "This low PSF before the head of the lahar arrives at each station could…"

**Line 340: When the DR in COLL is analyzed the authors refer to a phase change around 20 min after the arrival of the flow. Honestly, I am not able to visualize such a change in Figure 6. A similar change occurs around minute 12. Could the authors explain this confusion?** We are not referencing the exact time of the phase change, but rather the duration of the phase. We changed the wording of the sentence to emphasize this more: "The frontal surge or phase 1 of the lahar can be seen in the DR (Figure 6) as well. For every station along the channel the DR has a slight drop when phase 1 passes the recording station (Figure 6, dashed line). The elongation of phase 1 also has a correlation with distance from source, where the dip in the DR lasts for only ~1 min at RTMT, ~5 min at TRAN, and approximately 20 min at COLL.

**Reviewer 4:**

**Comments**

**I think the explanation concerning the orientation of the sensors (lines 235-238) should be moved to the section "Data". All seismologists are familiar with Z, NS, EW, or Z, R, T orientation/rotation, but may not know how a senor should be orientated in order to record signals due to mass flows.** Added "The seismic data were rotated to align with the channel in order to determine the differences in energy output between the flow parallel and cross-channel directions (e.g. directionality)." To the data section.

**In lines 273 and 275-278 some references are needed regarding the definition and usage of SCF and its spectral spread measure that is defined in equation 2**. References were added.

**In line 292 the authors write "dominate", but probably mean "dominant".** Changed

**I found it a bit hard to follow the description of the results in section 4.2 for the reason that it is based on the calculated metrics (DR, PSF, SCF), One of the main points of the manuscript is to illustrated the differences of mass flow induced seismic signals with distance from the volcano. I would therefore find it more natural to describe the results for each station used (RTMT, TRAN, COLL) possibly breaking the section into three smaller subsections.**
We separate the results section according to each station, but in section 4.2 we are discussing the evolution of the lahar signals. Due to this we find it better to discuss by lahar phase and how each phase changes with distance down channel rather than discuss each phase at each station separately. We have gone back through and tried to make the language less confusion so the audience can follow along easier. We also separated the section into 3 sub-sections to make the connections easier for the reader.

**In the discussion I found no mention on the application of machine learning algorithms that could help in the objective and real time identification of features compatible with mass flow induced signals. I suggest that the authors add a paragraph where they discuss such and application and how it can be implemented for the specific case of Mt. Ruapehu monitoring. I would like to note that a clustering algorithm could be applied to figure 8a in order to obtain a more objective set of clusters centroids instead of relying on visual identification.**
We have added a paragraph at the end of the discussion in section "Implication for monitoring". We discuss how the results of this paper could benefit and be a part of machine learning algorithms and implemented for real-time monitoring.

**I tend to agree with one comment from a previous reviewer that the results per station should be summarized in a table. This will help the readers remember easily the range of values of different parameters that are calculated, when going through section 4.2.** We have created a table with all the listed/calculated parameters in the manuscript. See table S1

**Reviewer 2:**

**Comments:**

**As I said before, the authors still give no information on where the data used in this study are located and how they can be accessed. This study is not reproducible without that, and most journals require that.** The data is in the Data Availability section

**Also as I said before, the authors still give almost no detail on how they did some of their analyses, so again this is not a reproducible study. The peak spectral frequency in particular has no details still despite both me and the other reviewer asking for that. The authors just say they compute a frequency spectrum and take the peak, but there are many ways of estimating a frequency spectrum with various ways of smoothing as well that might give different peaks. Did they just compute the FFT with no smoothing and take the highest value, or did they use welch's method, multitaper or some other approach that attempts to reduce the noise and stabilize the peaks? Same with directionality, they need to say exactly how the amplitudes used in those ratios are determined, did they just take the ratio of RMS amplitude for each component for each 10 s window or did they average ratios of envelopes from Hilbert transform? Was any filtering applied or is this all done with the full broadband data (hopefully they filtered because full broadband data can have a lot of very low frequency noise, especially in the horizontals)? No information at all is given on how the sediment concentration data was collected and very little on how velocities were estimated. All the details of the methods that someone would need to reproduce the study need to be stated explicitly in the text. Ideally they would have a methods section that details these things all in one place before the results rather than sprinkling these details throughout the text.** How each method was done is in the data or results sections. We added some additional information to please you.

**The abstract and introduction in particular contain imprecise and sometimes exaggerated language, but there is imprecise or casual language throughout. I made some line by line comments below but I didn't have time and it's not my responsibility to do this in great detail so I'd suggest the authors do a very thorough re-reading to improve the quality of the scientific writing in general.** See line by line comments.

**The "evolution of lahar signals" section on L313-L437 is still extremely speculative and also very hard to follow, my head was spinning trying to digest it. The authors did not respond to this comment that I also made last time and did not improve this section. The authors are, in my opinion, often overinterpreting the meaning of very subtle features of the seismic signals by invoking hypotheses from different past studies that are, in themselves, sometimes speculations. I think this paper would be much stronger if the authors focused on making their own interpretations that are supported by the other non-seismic data they have that tell them what the flow was doing over time and space for this exact event rather than speculating wildly on the meaning of seismic attributes alone based on findings from other studies of other flows elsewhere. The authors say there are videos, flow monitors, stage**

**height measurements, pore-pressure sensors etc. at 21 monitoring locations. No detail was given as to where these instruments were, but I presume they know how the flow behavior was evolving quite well over time at at least some of the seismic station locations. The authors should compare the time of those changes directly to the changes seen in the seismic data and use that for making interpretations just on this event. And then, those findings could be compared to some of these other studies to see if they are, in fact, also finding the same thing. That would provide a lot more scientific value than trying to overinterpret based only on what those past studies found. The comparison of directionality with sediment concentration is a great example of that, and I think it's one of the more interesting parts of the paper. More of that is needed and it seems like they should have the data to do that.**

We did respond to your comment, just because our comment is not what you determine to be enough does not mean it is not right. How is the signal elongating speculative? How is the DR and frequency changing speculative? The papers we cited in this section are based off of channel side seismic analysis of mass flows and rivers, so are you saying they are all wrong as well? You say "this paper would be much stronger if the authors focused on making their own interpretations that are supported by the other non-seismic data they have that tell them what the flow was doing over time and space" this is exactly what we are doing. Why would we use data from monitoring stations that don't have seismic data (you think we are speculating now, how about if we included other non-seismic data from >5km away, just think of the speculation?), we are not picking our data to fit a purpose. There were only three seismic monitoring stations as noted in the manuscript. You as in your other comments are twisting our words to set your agenda. As for the section 4.2 we have split it up into subsections to make it more easier to follow.

**Along the same lines, in the same section, L313-437, the authors still do not give any physical explanations for their numerous speculations like "a low PSF could represent the supercharged stream flow pulse" or "the ~10 Hz PSF may be explained by flow processes" to name two of many speculations made in this section. The authors responded to my comment on this last time by saying that they cite papers that describe the physical reason the seismic signals change, and that they describe these physical reasons through the manuscript many times. I am not satisfied with their response. First of all, the authors should not require their readers to read all the citations to understand the physical reasons why different relationships between seismic signal properties and flow properties might occur, they should paraphrase that relevant information in this text. And secondly, they do not actually explain the physical reasons anywhere in this section as they state in their response. Perhaps they didn't understand what I meant so I will try to clarify by giving just one example: on L339, the authors say "The reason the DR decreases during phase 1 for the 2007 lahar could be due to the parallel component being more sensitive to flow processes than bedload forces". But they don't say why this might be the case. That's what I'm asking for. So in this case, for the DR to be variable, it requires a change in the polarization and/or the type of seismic waves and/or the direction those waves are coming from, so why could the parallel component of a seismometer be more sensitive to flow processes? To explain the physical reasons behind it (and I'm not saying this is right), one could say something like, for example, that granular fluid flow induces more shear forces on the bed and side channel and that might generate**

**more SH waves that could convert into Love waves, and that might cause higher flow parallel amplitudes than cross-channel amplitudes when the highest amplitudes are coming from the part of the channel closest to the seismometer. Whereas for bedload, the impacts might be more vertical (assuming they just mean particles bouncing along in mainly fluid rather than bedload during a debris flow, which would have more shearing) and thus would generate more Rayleigh waves that might have a stronger cross-channel component if the signal were strongly dominated by energy coming from right next to the sensor and not other adjacent parts of the flow. The tricky thing with the elongated source of debris flows, and especially with stations being so close to the channel though, is that there is likely a lot of energy coming from all parts of the channel near the station and it may not always be dominated by processes happening right next to the station and that is probably muddying your directionality ratios, but I digress. In any case, explanations for why the seismic wavefield is altered due to these various explanations that are invoked is what I'm asking for. Give some logical explanations for why the invoked relationships between flow characteristics and various observed features of the seismic wavefield might make sense physically. What would be better is less speculation though, as described in my previous comment.**

If you would like us to write up a summary of each paper we cite, we would be happy to write a book instead of a research paper. The observations here are based off not only the seismic data but also the visual observations as well, as noted in section 4.2 and throughout the manuscript when referencing figure 2. From the observations we then go into the cited literature to find similar instances and then cite the paper and give a brief summary of what they observed or found as well. This is how a discussion is wrote. We never once said that this=that, if we were actually "discovering" a result, it would a have been in the "Results" section. It is funny that your amazing example of L339 where you claim we do not describe anything or "are speculative" or "not giving a physical reason" is actually explained by reading more. If you read before L339 we explain why in terms of frequency, and if you read after L339 we write "During phase 1 discharge increases, sediment concentration is low, and streamflow dominates resulting in a low DR." Furthermore, in terms of seismic directionality and forces, we describe that throughout and also list citations in which they have looked into that. It is outside the realm of this study on what the theoretical seismic wave propagation is from changing dynamics internally in the lahar. All your other comments about lahar signal sources and physical characteristics are all noted in the implications for monitoring section. In order to come to some kind of middle ground, in which we are trying to, unlike you, we took the advice of the other reviewers and have updated the discussion sections.

**While the authors did add an H/V analysis to investigate site effects, which I appreciate, they stuck it at the very end as an afterthought. It should have been when they introduced the frequency analysis in order to justify that their approach is valid before they start making interpretations. They also give conflicting information on how they did the analysis in the supplement. They said they used noise in the text and in the supplement text, but the label on the figure says they used the signal of the lahars.**

We disagree with this and think the placement of the H/V analysis is appropriate where we talk about implications for monitoring systems. There is no conflicting information, you read the

text wrong. We state that the background noise is streamflow because there is always streamflow in the channel (i.e. streamflow dominant).

**The authors added a new frequency estimation method, spectral centroid frequency, to address the concerns that myself and the other reviewer both had about the stability of the peak spectral frequency. I appreciate that, and I do appreciate that some details are given about how its computed, but it's also added late in the paper and only as a way to support the use of the other spectral metric. If this is more stable, why not just replace the peak spectral frequency with this one? Or at least put it on the same plot as the other frequency metric, it didn't seem to match up as well as the text leads one to think, the changes seem more subtle, but it's hard to tell without being able to actually compare the two more easily.** We do not replace the new frequency analysis with the peak frequency because overall the peak frequency analysis is easier and faster to process than the detailed spectrogram analysis (e.g. implications for monitoring). We also added a table in order to directly compare the results if you want to compare that way as well.

**Line by Line comments:**

**L13-14: This sentence doesn't make sense, the "changing landscape" implies long term volcanic processes but the part of the sentence saying it can "drastically transform the properties and dynamics of the flow" implies either changes that happen within the channel during a single given lahar due to erosion and deposition I presume or changes that happen along the course of the flow as it reaches lower slopes and wider channels etc. I'm guessing you mean the latter but I'm not sure. Consider rewriting for clarity.** How does this imply long term? Many "fast" volcanic processes can significantly alter not only the landscape but also the eco- and biospheres. Changed to "Monitoring for lahars on volcanoes can be challenging due to the ever-changing landscape along the flow path, which can drastically transform the properties and dynamics of the flow."

**L15-16: Odd English usage. The saying is "tailored to" not "tailored between" or "tailored at", so change "between" to "to", and change "but at" to "but to".** We are not referencing a saying. The sentence is grammatically correct. "tailored" just mean customized.

**L17-18: This seems like exaggerated language, I doubt any emergency manager would say the thing they need to understand first is how a lahar transforms over time. Tone down the language, don't use the word utmost, and find a more tempered way to say this is useful for hazard management. You could be more specific by saying how the lahar evolves influences how big it grows and how far it might go and what areas it might inundate or something like that.** Really? There are not hundreds on numerical modelers that model how flows evolve over time? How about observatories, you don't think they would like to know how a mass flow or a

flood evolves overtime? Ok emergency managers, you don't think they need up to the second information on how a hazard changes to determine what there next steps are?

**L26-27: Used by who? Use active voice. I'd also say how these approaches were used, e.g., something like "We analyzed 3-component seismic amplitudes, frequency content…to investigate how the evolution of the lahar's behavior is reflected in the seismic wavefield." Also say what directionality is very briefly in the abstract, (cross channel amplitudes/parallel amplitudes) since it's not a standard seismic parameter.** Again, this is your personal preference. We clearly state what methods data we are using in this paper and what we found.

**L31-33: It would be incredibly hard to understand what you mean by this without reading the entire paper in detail first. I suggest being more specific about what the pattern is on each of the three stations. Also, why are you not mentioning the sediment concentration part of the study in the abstract? I found that to be one of the most compelling and novel parts of the paper.** How is this hard to understand, please explain? The sediment concentration is used only for one station for only a small fraction of the lahar. You would be correct if this was a study on the comparison between seismic signals and sediment concentration or if we had data from all three sites.

**L33: Extraordinary promise is exaggerated, I suggest removing the word extraordinary. I also haven't seen in this paper any specific suggestions of how directionality ratios could actually and practically be used in monitoring so this statement isn't really reflective of the paper.** How is this exaggerated? Seems like you are against the idea of directionality ratios. You would not want to use directionality ratios on streamflow channels? We show there is a large change in DR when the lahar arrives, is that not good enough of a signal for you?

**L40-41: This sentence and the rest of the paragraph seems specific to debris flows/lahars, but the way this paragraph is written, you're still referring to all volcanic mass flows. Why not just start this paragraph specifically talking about just lahars and forget about mentioning the PDC's and debris avalanches?** PDC cannot move debris great distances or are a threat? So you are saying we should not study PDCs or debris avalanches with geophysical data?

**L42: Larger than what? And what kinds of changes?** Did you not read the sentence before, clearly larger than other flows. Again did you not finish reading the sentence…" changes to the landscape and surrounding ecosystems"

**L44: A flood can't "have" a warning. "outburst floods can occur with little or no warning". Really?** There are a lot of papers I could send you and news articles that say different.

**L45: Eruption sources of what? "Lahars triggered by eruptions can be anticipated…"** Changed to "Eruption triggered flows"

**L46: various methods is too vague.** Nice job nit-picking this sentence, because if you read the rest of the sentence you would see all the methods nicely wrote out.

**L48: How is "the intensity of rain" a technique? Setting alarm thresholds based on the intensity of rain would be a technique.** Added the word "measuring"

**L52: change forestry to forests.** The correct word in this context is forestry.

**L56: I suggest replacing "predict and investigate" with "characterize".** Predict and investigate are better for the meaning of this sentence.

**L57-58: How can this be the first step? People have been doing this for many many decades, and the next paragraph starts off talking about some of that work. This also implies that we don't know much but we know a lot.** We are not describing past work here. The "first step" is the first step every time. How else would you make progress on a subject? How does this imply anything? In the next paragraph, we even acknowledge and cite many people and methods from the past, how is this in anyway saying we are first and only people who have worked on flows?

**L59: Specify that you are talking about in-situ methods here.** Added

**L61-63: References needed.** Added.

**L66: The paper cited here as showing that seismometers can be "capable warning systems" isn't even about an actual warning system.** How is using geophones to turn on flashing lights when a debris flow passes by not a warning system?

**L66: I would disagree that they can "accurately" estimate flow properties, seismometers are quite inaccurate compared to other methods (depending on what flow properties & flow dynamics, it'd help if you were more specific about what you mean by flow properties) and prone to many limitations that aren't really discussed here**. Please read the citation provided. Also thanks for pointing out that physically measuring the flow properties will be more accurate than using a non- in-situ sensor. We are stating that geophysical instruments are good for measuring at a safe distance.

**L68-70: This sentence is in direct conflict with the previous one that said geophysical instruments could already provide accurate estimates.** How? We never stated that these methods were perfect

**L71: Using all three components isn't a "technique." Rephrase to say something like "Using techniques that use information from all three components…" or similar.** Using all three components is a technique.  Technique- "A way of carrying out a particular task." Using all three components sure seems like a way of carrying out a particular task.

**L72: Again, using all three components isn't a technique in itself and can't characterize anything by itself. Using all three components to do what?** Read the sentence, we literally give examples. See comment to line 71 as well.

**L77-79: Provide a brief explanation here about what directionality analysis is, because it's not a common term. The next sentence sort of implies what it is, but just say it directly. Also provide a bit of explanation for how it could provide the information listed. i.e., why would the polarization of the seismic waves be sensitive to changes in those things.** Inserted definition of directionality.

**L86: Change "entrapped" to "entrained".** Changed

**L88-89: As I said before, please give a few words explaining what you mean by laminar or plug-like. It isn't enough that they are in the references already, paraphrase here. I am specifically asking for this because later in the text what you talk about as plug-like or laminar isn't what I had in mind so you need to be sure everyone knows exactly what you mean.** Added "(limited internal motion and collisions)"

**L93-107: I don't see why it's relevant that Cronin et al had three models at different distances, the differences between those three models aren't described here in any meaningful way. Just stick to the four phases. Also, it would be helpful to include here what part of this sequence you are referring to as the lahar "head". That term is used later in the text and I don't know what part you mean by that exactly or what characteristics it has.** The whole paragraph describes the phases of the flow. Added text throughout to help with the terminology.

**L122: change "entering" to "to enter".** Changed.

**L125: Are there some missing words here? How can landslides be "along with" a channel? Perhaps you mean landslides contributed additional sediment to the channel?** Again you are not reading the sentence. We are noting that there were landslides as well. We changed the wording of the sentence to make it easier to read.

**L131: change "thus an" to "thus had".** Added "contained"

**L132: How can a debris flow be filled? Suggest changing "sediment-filled" to "sediment-rich".** Changed to "laden"

**L133: I flagged this sentence last time and it still makes no grammatical sense. Please fix. Also, provide detailed information on how velocities were estimated before presenting velocity results.** Changed to "At ~8.0 km from source, the lahar velocity was recorded at ~ 9.5 m/s and had an estimated 6 m of downcutting, showing the capability of the lahar to deposit and erode massive amounts of material". Again how the velocity was measured is stated in the data section, just need to read.

**L141: Awkward transition, the way that it's worded sounds like you're talking about another lake-breakout than in the previous paragraph. Suggest changing "properties of a" to "properties of this" and maybe other changes to make the transition smoother.** This is a

overview paragraph, almost every published paper has one at the end of the introduction. We rearranged the first sentence to make it read easier.

**L153: change "sampling" to "sampling rate".** Changed.

**L154-155: This is not accurately described. To rotate to flow parallel requires both the North and the East component, you can't rotate just the North component to get flow parallel as this phrasing implies. I suggest just saying you rotated them to align with the flow parallel (P) and cross channel directions (T).** How does it imply what you state. The sentence says "the recorded data were rotated to align North as flow parallel (P) and East as the cross-channel direction (T)" We never state that the North was used to convert the East component.

L155-156: **This is an awkward way to say the seismometers were installed to be level. Broadbands have to be installed level anyway, not sure why this sentence is necessary or why you can't just say the sensors were leveled**. We deleted the statement.

**L164: Arrival times of just the flow front or of different phases that passed? Also, discuss the temporal accuracy, especially of the eyewitness reports which are notoriously not accurately timed.** We stated that arrival times are based off both images and eye witnesses.

**L165: Provide details on how flow velocities were estimated from imagery and some more info about the flow meter that was installed at TRAN. Do you have continuous flow velocity estimates at TRAN? If so, why aren't these used in the paper? Also, this information should be before flow velocities are reported.** The velocity data is not presented in this manuscript because it does not add any additional information needed.

**L167: Insert methods section here.** We discuss how each method is conducted where it is introduced and give a brief overview in lines 168-174.

**L171: Delete the word amplitude after PSF, you are not reporting the amplitudes of the frequency in this study as far as I can tell, just the frequency.** Deleted.

**L172-174: This kind of thing should be labeled on the figure and in the caption, not necessary in the main text.** This is labeled on the figure, look again

**Figure 2: Give time stamps consistent with your figure time axes for the photos, especially for b,c,d,e. Possibly also label them on the figures. What is a "low PSF beginning of a lahar body"? Use more precise language.** Added extra language to explain each phase better.

**L183: Spectra should be changed to spectrum if you're talking about just one. Also, more detail should be given as there are many ways to get a frequency spectrum. I'm guessing you're talking about computing the raw FFT and taking the absolute value but I shouldn't have to guess.** Added.

**Figure 3-5: Label the panels with letters and then use those to refer to them in the text instead of referring to them by the color of the dots. I'd also suggest labeling the arrival of all the lahar features you talk about in the text that you know about from the videos etc. on these figures. It would greatly help people follow what you are talking about.** Added labels.

**L203: Define what you mean by lahar head here or earlier in the text.** Completed.

**L205: Mention upward sweep of vertical frequencies?** The increase in frequencies is mentioned in the text.

**L224: What is the difference between the lahar front mentioned here and the lahar head mentioned earlier? Be sure all the terms you use are very clearly defined early in the text.** We have defined these throughout. Also have added more definitions in the introduction.

**L232-234: Here or earlier such as in the introduction, talk about why directionality could potentially contain useful information, physically. i.e., why would some directions have stronger energy than others as the flow behavior progressed? What is changing that is changing the seismic wavefield?** Again if you read the rest of the paragraph you would have found this information.

**L244-246 sort of starts to scratch the surface, but that should be earlier in the text and also you need to get into why, physically, would the directionality of seismic waves be influenced by those flow properties.** This is not a theoretical seismology paper. We note some theories and cite others that describe why directionality might occur here and later in the manuscript.

**L239: You should, however, discuss some caveats of this, including the fact that you will have energy coming from a large section of the flow that is more altered by attenuation, not just the portion right next to the station.** We discuss this in the discussion section.

**L240-241: Give details on how exactly you estimated the amplitudes for this, ideally in a methods section earlier (amplitude method, filtering, duration of windows etc.). I can piece together clues sprinkled through the text to guess what you did, but again, I shouldn't have to guess. Just explicitly say the methods up front in enough detail that someone could do exactly the same thing you did.** We do all of what you are asking for on lines 168-175 and lines 272-274, you must have forgot to read again. To make up for this we have restated all the information in section 3.2

**L243: The paper cited here was not about a warning system so saying DR was used for warning purposes is misleading**. Deleted warning system

**L253: TRAN did not behave similarly to RTMT as this sentence implies, TRAN started low and stayed low until 10 min. RTMT started around 1 and barely dropped before increasing again. Consider rephrasing.** You might want to read and look at the figure again. Both were below one, then went above one, then went below again.

**L271: A frequency can't describe lahar dynamics. Consider rephrasing maybe something like to understand "if changes in PSF are related to changes in lahar dynamics"?** There are a lot of papers I could sent you that say differently.

**L272: What is meant by frequency constraints is not clear. I assume you mean the stability of the frequency peaks needs to be analyzed? Consider rewording.** We considered it and constraints is ideal.

**L274-275: Normalizing by what? By the peak amplitude? Total energy?** Added definition for type of normalization.

**Figure 7: This plot needs to also have the context of the flow arrival times on it and the other frequency estimates. It's also hard to compare to the other plots when it's by itself like this**. All the data on the figure serves a purpose. If we add everything you want the figure will become very messing and hard to understand.

**L296-297: These features need to be labeled on the plot and/or times of these features need to be given, ideally both. I have no idea when these things are happening on those plots.** See comment for figure 7.

**L300: What is "this" that explains the bimodal distribution? I'm guessing you're implying that the peak spectral frequency is jumping for TRAN because there is energy in two bands and they are similar in amplitude so which one is higher and thus serves as the PSF varies slightly over time. Be more explicit, again I shouldn't have to guess at what you mean.** Again if you read the sentence before you would understand. It is called a continuation of thought/transition.

**L326: Do you know from your videos or other data that the supercharged stream pulse is passing at this point? I feel like with the amount of data available for this study, speculation about what is happening at various times during the signal is not needed and instead the authors should rely on what they know from observations directly from their non-seismic data instead of speculation based on other studies. I could make this comment several times in this section, but I won't point them all out as this whole section needs a reworking based on my main comment above.** This is exactly what we are talking about. We are discussing what could be causing the changes in the frequency. We know there is a streamflow pulse or the uplift of streamflow water from the underflow of the lahar. We are not just throwing out every random idea and hoping one sticks with the reader. All the data and images show there was a pulse. We are discussing the link from the visual/seismic amp to PSF response. We added a reference to figure 2 to help the readers.

**L331-332: I don't understand the logic of this statement, or why a low frequency zone suggests lahar elongation. Please clarify.** What do you not understand?, the low frequency

zone becomes longer at each station the further away from source. Again please read the entire sentence.

**L333: The term "flow processes" could mean pretty much anything. What else is there?** Again read the full sentence.

**L334: I don't understand what this means, a flow is discharge, how can it be more sensitive to discharge at one time than another? Do you mean the seismic signal is more sensitive?** There are a lot of process that occur within a flow. Some have stronger signals than others which can described as being sensitive to a particular dynamic. We never stated that the flow did not have discharge, we have no clue where you came up with that.

**L338: Why is phase 1 not labeled on the plots? Also, do you have confirmation from your other data when phase 1 is passing? If so, is it the same time period as when DR is behaving this way? See main comments.** We describe the phases in detail in the text. We have also referred to figure 2 multiple times and have added more to point out the change in imagery as well.

**L342-343: Do you know this is the case from your data from this event?** This is based off of past models. Again we never state this as fact. This is a discussion of the observations we made.

**L350-351: What else is there besides turbulence or sediment transport? This doesn't seem very meaningful. Also, why would one expect the frequencies to be the same between different monitoring locations? Frequency is very dependent on distance from the source, material properties, among other factors.** We never stated they are the same. If you are referring to the cited material and the > 30 Hz for channel side stations, this statement does not emphasize an exact frequency does it?

**L354: Bulking up into a full 4-phase lahar doesn't make sense to me. The way the 4-phase lahar is described earlier is an evolution over time and distance, not four phases that happen all at once as this implies. "Bulking up" seems to me like something that happens during that process, in phase 2 and 3, not all at once. Do you mean it has reached phase 3 at this site?** How does this sentence imply all four phases are occurring at once? We never state they are occurring at once. We are describing the evolution of the lahar and its ability to bulk up and form 4 phases.

**L367: Same comment as above. Do you mean it reached phase 3 or 4 at this point in space and time?** At TRAN the lahar was bulked up and all four phases of the flow could be seen passing the monitoring station. Again we never state that it is only one phase or they happen all at once anywhere in the manuscript.

**L373: This is the first time relative amplitudes of vertical vs. horizontal have been mentioned. If this is something of interest in this study, it should be presented in the results section. I also feel that this statement is made as if it is always true, I'm sure there are other reasons that**

**vertical amplitudes could be higher than horizontal. It also probably varies with frequency.** The data is in each of the figures and RMS amplitudes are referenced in the results section. How is it presented as "always true"? we cite two papers and use the word "note". We never state this as fact.

**L375: Bimodal pattern of what? I assume you mean of the peak frequency? Say so explicitly.** Added "frequency"

**L387-389: The wetted perimeter was not examined or even discussed thus far in this study, so how is the reader supposed to know that an increased DR accompanied an increased wetted perimeter for this lahar? Where is this data?** We have rewritten the section on this to refer the reader to figure 2 and the images from TRAN.

**L394-395: Here is one instance where a reason for a pattern is given, this is good, but is incomplete because it doesn't say why lateral excitation relates to the DR.** Added "increasing the cross-channel signal"

**L397: I would disagree that this figure shows a "good" correlation. It looks very scattered. Did you compute the Pearson's correlation coefficient?** Changed to "the slight increase in DR overall when the PSF increases"

**L398: What timeframe? Also, Figure 2f looks like laminar fluid flow from what I can see, I don't see plug-like flow. To me, plug-like flow involves a very high sediment concentration and lower pore pressures and is like a big plug of sediment that is being kind of bulldozed along by a wetter flow behind it. Plug-like flow seems like it would have a lot of particle collisions, but this sentence implies it has fewer.** This is a hyperconcentrated flow not a debris avalanche or even a debris flow. Your definition might work with a rolling grinding front in a dry channel instance, but not here.

**L403-404: Same comment as for L397.** Added R2 value

**L414: I don't know what you mean by "traditional" here and elsewhere. Be more specific. Do you mean it reached phase 3 of the 4-phase model you are invoking? Do you know that these descriptions are the case at that time from video or other data or is this based just on the seismic interpretation?** Again please read the manuscript properly, we describe this throughout. Added "4-pahse".

**L443-444: The only actual measurable properties that were estimated and presented in this study came from non-seismic instruments (velocity and sediment concentration), is that what you are referring to here?** We are talking about exactly what we say

**L446-448: Wasn't this shown by Procter et al. and other references already?** Can we not show it as well? Is that forbidden? And no you are wrong the other papers only presented work on the first < 10 km of the flow.

**Figure 9: Should the x axis be labeled as time since this is showing the lahar passing by fixed points? Also, why is PSF not also L for the bow wave in panel C as it is for the other two? Also, say what the values ranges of PSF and DR are for D, I, H, L, and M**. This is a conceptual model it does not need axes. It is mixed for the bow wave because the frequencies in each component are different. The values are in the table 1.

**L472: what is considered "when the lahar arrives"? The lahar head? SLF/HF? TF?** Added "head of lahar"

**L477: If new flows are going to be used in this paper, you need to provide more background information about them.** Read the paragraph. We state where you can find details on these flows. We even state how these lahars occurred and when and where they flowed to and from.

**Figure 10: Show when each lahar has passed/ended because in the text you refer to times when the lahar has passed and talk about things that happen for the "entirety of the event" but do not say what time that includes.** There are dashed lines on the figure that show this.

**L485: What observations? No references were given for the statement this is referring to.** Added citations.

**L493: Can you be more explicit about what you mean by little evidence? I do see elevated seismic amplitudes at these times, they just aren't as high as at later times.** Rewrote sentence.

**L505-507: The amplitude differences weren't really examined in this paper, and certainly weren't compared to any data on bedload. This is also stated as fact and I'm not sure it's established scientific fact, even if there is a reference.** We did not state this as fact, again please read the sentence, we use the word "discoveries" not science fact. Amplitudes are a part of this manuscript and are shown in most figures along with PSF and DR. This sentence/paragraph is also talking about the importance of using all three components. Added the word "may" so you do not get confused.

**L511-515: These sentences are all way too vague.** How are they too vague, we are listing reasons for concern in the results, hence the discussion section, hence the section in the discussion section called implication for monitoring. It is outside the realm of this paper to conduct tests on every one of these concerns. We state that these are issues that must be known/dealt with in the future.

**L519: I don't see the logic behind why a flow overtopping the channel would reduce the amplitudes.** No channel= less cross-channel noise from collisions.

**L522: I don't recall hearing about the flow breaking out of the channel at COLL previously, are we supposed to know about this already?** True we don't ever mention that, Again please read the sentence, we specifically state TRAN

**L525: Awkward description, please describe more succinctly**. The sentence is grammatically correct.

**L539: RTMT and COLL do have distinct H/V peaks, this is not an accurate statement. The smoothing level you used in your H/V analysis also controls how narrow the peaks can be.** No they do not, we guess we will have to agree to disagree on this one.

**L560-561: This sentence implies you know how the flow evolved with distance only from the seismic sensors, but you had several previously published papers on these flows and other data as well. This is misleading.** How is this misleading, we show in the manuscript these signatures in the seismic data and then back them up with other data. Again how are we implying? "yielded an understanding" wow we sure implied there.

**L568-569: I didn't see anything in the paper that explained how a DR could be used in a warning system explicitly. Some discussion is needed for what exactly they would be used for in the warning system and how, and what additional analysis is needed to make that possible.** See new paragraph at end of discussion.

**Supplement:**
**Figures S1-S3 Label times of different arrivals on these plots.** See comment to this same question above.

**Kommo should be Konno.** Changed.

**What was the bandwidth of the Konno and Ohmachi smoothing?** Added "and a bandwidth coefficient of 20"

**In H/V analysis section, say how much data was used. Was it the entire day?** "where ambient noise (streamflow dominant) from the previous day was split up into 10 s time windows with 50% overlap." Again please read.

**Frequency is misspelled in Figure S4 labels.** Fixed.

**Figure S4 says the H/V results are for the lahar signal, but the text says ambient noise from the previous day was used**. Read again please, we never say that.